# OPENHELIX: EMPIRICAL ANALYSIS OF DUAL-SYSTEM VLA MODELS FOR ROBOTIC MANIPULATION

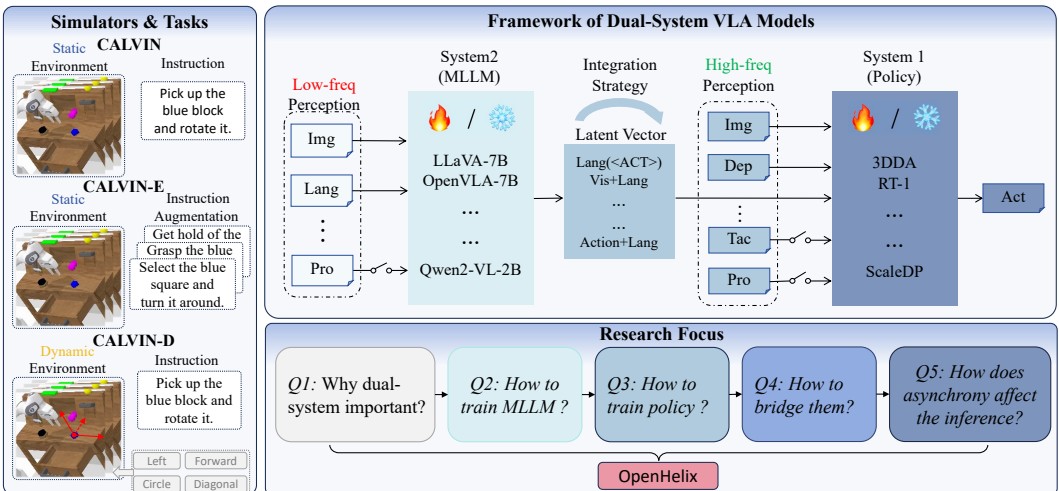

Figure 1: Overview of this work. We present a comprehensive study of dual-system VLA architectures. To more rigorously evaluate the effectiveness of this design, we introduce two augmented benchmarks: CALVIN-E and CALVIN-D. Our investigation addresses five key questions: the necessity of a dual-system, strategies for training the MLLM and the policy, approaches to bridging the two components, and the impact of asynchronous inference on performance. Building on these insights, we propose a simple yet effective dual-system VLA model, OpenHelix.

## ABSTRACT

Dual-system vision-language-action (VLA) architectures are emerging as a promising approach in embodied intelligence. However, current works lack consistency in training and evaluation protocols across high- and low-level modules, making systematic comparison and rigorous analysis challenging. In this work, we conduct a comprehensive study of core design principles in existing dual-system VLA architectures and introduce DSVLABench, a new suite that covers diverse evaluation scenarios and standardizes the assessment pipeline for various architectures. Our results show that prompt tuning preserves multimodal large language model generalization, fine-tuning from pre-trained policies outperforms training from scratch in policy learning, and pre-aligning projectors with auxiliary dynamic visual tasks significantly enhances latent space training. Additionally, we find that the frequency of high-level updates has minimal impact during asynchronous inference, with latent embeddings remaining robust to dynamic changes. We hope our findings provide practical guidelines for developing more generalizable and robust dual-system VLA models.

## 1 INTRODUCTION

Vision-Language-Action (VLA) models (Wu et al., 2024; Kim et al., 2024; Black et al., 2024; Ding et al., 2024; Wen et al., 2025b; Song et al., 2025; Zhao et al., 2025a; Zhang et al., 2025; Zhao et al., 2025b), which are co-fine-tuned on large-scale robotic trajectories and Internet-scale vision-language datasets, demonstrate impressive generalization to novel objects, diverse instructions, and

emergent behaviors. However, their considerable model sizes present significant obstacles for real-time deployment (Han et al., 2024). For example, the 55B and 5B variants of RT-2 operate at only 1–3 Hz and approximately 5 Hz, respectively (Zitkovich et al., 2023), while lightweight models such as BC-Transformer can achieve inference speeds of around 50 Hz (Mandlekar et al., 2022). In addition, pre-training and fine-tuning on embodied data are both computationally expensive and prone to issues such as domain shift and catastrophic forgetting. As a result, applying large vision-language models in robotics—while maintaining robust multimodal understanding and low-latency control—remains a fundamental challenge that demands careful architectural trade-offs.

To tackle these challenges, dual-system VLA models (Shentu et al., 2024; Zhang et al., 2024; Han et al., 2024; Bu et al., 2024; Wen et al., 2025a) have been proposed, inspired by dual-process theory (Tversky & Kahneman, 1974; Kahneman, 2011; Evans, 2008; Neys, 2006), which posits two distinct cognitive systems. System 1 is fast, automatic, and intuitive, analogous to lightweight policy networks that are highly efficient yet task-specific. System 2 is slow, deliberate, and general-purpose, analogous to large models such as multimodal large language models (MLLMs) and VLA models. Rather than synchronously cascading these two systems, dual-system VLA models decouple their update frequencies: System 2 updates infrequently to deliver high-level decisions, while System 1 updates rapidly to produce low-level actions for real-time control.

Despite these advances, current benchmarks (Mees et al., 2022; Liu et al., 2023a) are not well aligned with the dual-system VLA paradigm. They often fail to disentangle the contributions of fast-reactive control and deliberate decision-making, or to evaluate capabilities such as language-conditioned generalization and adaptation to mid-task goal changes. To address this gap, we introduce CALVIN-E and CALVIN-D, two benchmarks built upon the contemporary robotic simulation platform CALVIN (Mees et al., 2022). CALVIN-E focuses on language-conditioned generalization in static environments, emphasizing the high-level semantic reasoning associated with System 2. CALVIN-D, in contrast, features dynamic task changes that require coordination between both systems for reactive adaptation and deliberate planning. Together, these benchmarks provide a principled testbed for evaluating the robustness, flexibility, and coordination abilities of dual-system VLA architectures.

Current dual-system VLA models vary significantly in both architectural designs and information flow strategies, as shown in Figure 1. These differences complicate direct comparisons and make it challenging to assess the effectiveness of individual components. To address this, we propose a comprehensive evaluation platform DSVLABench that examines five critical aspects: the necessity of dual-system architectures for vision-language-action task, training strategies for both MLLMs and policies, methods for learning and communicating latent representations, as well as asynchronous training and testing strategies. By evaluating these aspects across the CALVIN, CALVIN-E, and CALVIN-D benchmarks, we aim to provide a systematic comparison of existing architectures, offering valuable insights into their relative strengths and weaknesses.

Our non-trivial and insightful findings can be summarized as follows:

(1) The dual-system VLA architecture plays a crucial role, as performance degrades significantly for pseudo dual-system variants in dynamic scenarios.
(2) For MLLM training, prompt tuning is highly effective in preserving the model's ability to generalize to diverse textual instructions.
(3) In policy learning, fine-tuning from a pre-trained policy consistently outperforms training from scratch.
(4) In latent space training, pre-aligning the projector is essential, and introducing auxiliary tasks correlated with dynamic visual cues substantially improves performance.
(5) During asynchronous inference, the frequency of high-level updates has minimal effect on task success, and latent embeddings reliably capture static semantic features while remaining robust to dynamic environmental changes.

## 2 RELATED WORK

**End-to-End VLA Models.** End-to-end VLA models can be categorized into generalist and specialist approaches. Generalist models, such as RT-X (Brohan et al., 2023a) and OpenVLA (Kim et al., 2024), rely on massive parameters and large-scale cross-task datasets to achieve strong task adaptability and

cross-domain transfer, but their high computational cost and latency hinder real-time deployment. In contrast, specialist models (Chi et al., 2023; Ze et al., 2024; Prasad et al., 2024; Ke et al., 2024; Shridhar et al., 2023; Goyal et al., 2024; Fu et al., 2024) adopt lightweight architectures tailored to specific tasks, offering higher efficiency, precision, and responsiveness, yet they lack broad generalization. Compared with these end-to-end designs, dual-system VLA architectures aim to strike a balance by combining the broad reasoning capability of large models with the efficiency and stability of small task-specific policies.

**Dual-System VLA Models.** Recent works have adopted dual-system architectures that separate high-level reasoning (System 2) from low-level control (System 1). LCB (Shentu et al., 2024) employs LLaVA (Liu et al., 2023b) to produce latent goals that guide a 3D Diffusion Actor (Ke et al., 2024). DP-VLA (Han et al., 2024) grounds this paradigm in dual-process theory, where System 2 (e.g., OpenVLA (Kim et al., 2024)) extracts latent task representations to inform a Transformer-based System 1. HiRT (Zhang et al., 2024) utilizes InstructBLIP (Ouyang et al., 2022) with MAP pooling to obtain MLLM features that condition an EfficientNet-based policy. Robodual (Bu et al., 2024) further integrates OpenVLA with a ViT encoder and a diffusion transformer (Chi et al., 2023), enabling multimodal sensing and task-aware action generation. Overall, these methods share a common principle: leveraging high-level representations from System 2 to enhance generalization and control in System 1. However, they differ substantially in their concrete implementations, as System 2 may be an MLLM or VLA and System 1 may use transformers, CNNs, or diffusion policies, and no consistent paradigm has emerged. This highlights the limited understanding of dual-system architectures and motivates the need for a systematic empirical study, which we address in this work. A more detailed summary and comparative analysis about dual-system VLAs can be found in Appendix A.

# 3 BACKGROUND

**Problem Formulation.** The goal of vision-language-action model is designed to mimic demonstration trajectories in the format $\{l, (o_1, a_1), (o_2, a_2), ...\}$, where $l = \{w_i \in \mathbb{R}^d\}_{i=1}^{N}$ represents a task-specific language instruction of length N with an input dimension d, and $o_t$ and $a_t$ denote the visual observation and corresponding robot action at each timestep $t$. The input observation $o_t$ consists of multiple images from different viewpoints. The output action $a_t$ defines the end-effector's pose, which is decomposed into 3D location, rotation, and gripper state (open/close): $a_t = \{a_t^l \in \mathbb{R}^3, a_t^r \in \mathbb{R}^6, a_t^g \in \{0, 1\}\}$.

**Dual-System VLA.** The dual-system VLA model consists of two components: System 1 and System 2. System 1 is typically a lightweight policy network $\pi_\theta$, while System 2 is usually a multimodal large language model (MLLM) $f_\phi$. The System2 processes language instruction $l$ and observation image $o_t$, outputting the latent embedding $z_t$ for System 1. The System1 takes as input the conditioning information from the environment observation $o_t$, the latent embedding $z_t$, and proprioception $c_t$ of timestep $t$, predicting the action trajectory $\tau_t = (a_{t:t+T}^l, a_{t:t+T}^r)$ and binary states $a_{t:t+T}^g$ at each timestep $t$, over a temporal horizon $T$. Notably, both System 1 and System 2 must receive observation inputs $o_t$ independently so that their inference processes can be executed asynchronously.

# 4 DSVLABENCH

Previous research lack a fair comparison of dual-system VLA models due to varied training and inference methods. To address the lack of fair evaluation, we propose **DSVLABench**, which offers standardized pipelines for dual-system VLA models, enabling fair analysis of their generalization in language and dynamic scenarios.

**Simulators.** To ensure fair and meaningful comparisons with non-open-source models that have reported results, we adopt evaluation environments consistent with prior literature. Specifically, in accordance with LCB (Shentu et al., 2024) and RoboDual (Bu et al., 2024), we designate the **CALVIN** as our primary simulation platform.

**Evaluation Setting.** The canonical evaluation scenario features static objects and standardized language instructions. Nevertheless, dual-system architectures are intrinsically designed to synthesize the language generalization capabilities of large-scale models with the low-latency, high-frequency control afforded by smaller models, especially in dynamic environments. Accordingly, we perform

supplementary evaluations in two extended scenarios: **(1) CALVIN-E**: To assess language generalization, we employ enriched and diverse language instructions generated by GPT-4 (Achiam et al., 2023). **(2) CALVIN-D**: To examine robustness in dynamic cases, we introduce four distinct object movements during grasping tasks, as illustrated in Figure 2. Ablation experiments use 100 trials for efficiency, while the final evaluation in Table 7 and 9 uses 1,000 trials for comprehensive results.

## 5 EXPERIMENTS

Current dual-system VLA models exhibit substantial variation in both the selection and training paradigms of key components, including base multimodal large language models (MLLMs), downstream policy architectures, latent representations, and asynchronous strategies for training and inference (Shentu et al., 2024; Han et al., 2024; Zhang et al., 2024; Bu et al., 2024; Wen et al., 2025a). To facilitate fair comparison and systematic evaluation of design choices, we formulate five research questions to guide our experiments, explicitly considering practical constraints on computational resources. Specifically, we investigate the following questions:

**Q1:** Why dual-system architectures is necessary?
**Q2:** What training strategies are most effective and efficient for the MLLM component in System 2?
**Q3:** What training strategies are most effective for enabling policy learning in System 1?
**Q4:** How should latent representations be trained and communicated between the two systems?
**Q5:** How do asynchronous inference strategies influence system performance?

### 5.1 LIMITATIONS OF PSEUDO DUAL-SYSTEM VLAS? (Q1)

Systems such as RoboFlamingo (Li et al., 2024), $\pi_0$ (Black et al., 2024), and GR00TN1 (Bjorck et al., 2025) integrate a high-level MLLM with a low-level action expert. However, these architectures cannot be strictly classified as dual-system VLAs, as their action experts do not receive real-time perceptual feedback. Therefore, we refer to them as pseudo dual-system VLAs. While this design performs adequately in relatively static environments, the absence of real-time perception leads to significant performance degradation in dynamic settings. Below, we analyze the limitations of pseudo dual-system VLAs.

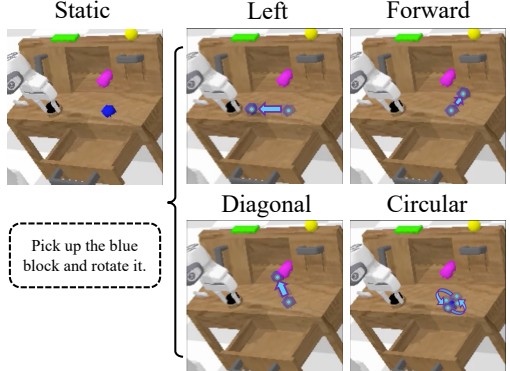

Figure 2: Four types of dynamic objec movement patterns on **CALVIN-D**.

**Experimental Setup.** We train models on the standard ABC dataset and evaluate them on **CALVIN-D** over 100 trials. The "Static" condition involves stationary objects, while "Left," "Forward," "Diagonal," and "Circular" denote four dynamic object movement patterns, as shown as Figure 2. Detailed results are provided in Table 1. Here, we select RoboFlamingo (Li et al., 2024) as a representative pseudo dual-system VLA.

**Analysis.** The RoboFlamingo (Li et al., 2024) model fails entirely in dynamic scenarios, achieving a 0% success rate across all movement patterns. This degradation stems from its LSTM-based policy, which relies on latent representations of past frames while lacking real-time visual feedback. While these representations are stable during training on static scenes, they become highly unstable when objects move during testing, causing a significant distribution shift and resulting in failure. Thus, to enhance the performance of pseudo dual-system VLAs in dynamic tasks, it is necessary to incorporate real-time visual inputs within a genuine dual-system architecture.

Table 1: Success rates on the CALVIN-D benchmark. Best results are shown in bold.

| Model | Static | Left | Forward | Diagonal | Circular | Avg. |
|---|---|---|---|---|---|---|
| RoboFlamingo (Li et al., 2024) | **100** | 0 | 0 | 0 | 0 | 20 |
| 3DDA (Ke et al., 2024) | 82 | **84** | **46** | **67** | **80** | **71.8** |

## 5.2 STUDY ON MLLM TRAINING STRATEGY (Q2)

In recent research like LCB (Shentu et al., 2024), HiRT (Zhang et al., 2024), and Robodual (Bu et al., 2024), their upstream large models are all fine-tuned. By contrast, although GR00TN1 (Bjorck et al., 2025) does not employ a dual-system architecture, it achieves strong performance under a frozen paradigm. The training paradigm of MLLMs is particularly crucial for preserving the generalization ability of high-level modules. To investigate this, we conduct experiments using both fine-tuning and frozen approaches.

**Setup.** For fair comparison, we adopt the LCB structure (Shentu et al., 2024): an LLaVA backbone (Liu et al., 2023b) is connected to the downstream policy via a newly introduced <ACT> token, with a CLIP loss (Radford et al., 2021) used to further align this token with downstream instructions. The downstream policy is consistently fine-tuned across all experiments. Based on this, we conducted experiments on the CALVIN benchmark to compare the effects of fine-tuned/frozen VLMs and the use of CLIP loss.

**Analysis.** As shown in Table 2, when the MLLM is frozen, the inclusion or exclusion of CLIP loss has negligible impact on performance. This is expected, as the CLIP loss merely adapts the fixed MLLM outputs to the downstream policy, resulting in limited effect. In contrast, when the MLLM is fine-tuned, CLIP loss becomes critical. Without it, fine-tuning may disrupt the alignment between conditioned inputs and the policy's attention mechanism at the initial training stage, potentially degrading performance due to a loss of semantic consistency across modalities.

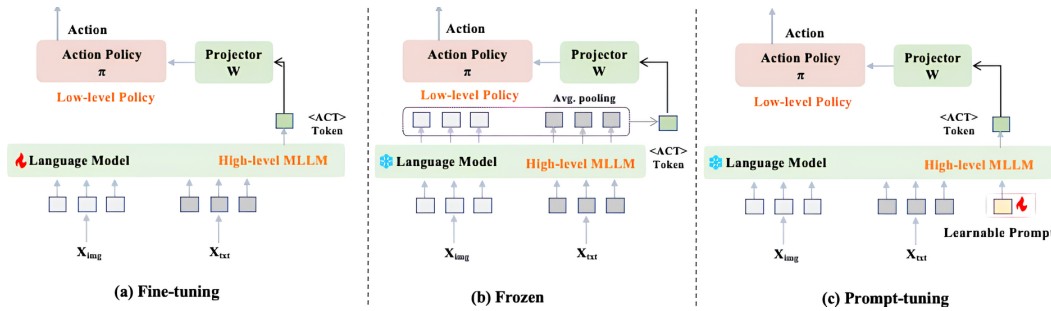

Figure 3: Three Different MLLM Training Strategy.

Table 2: Success rate comparison of different training strategies for the high-level MLLM on the CALVIN benchmark. FT denotes Fine-tuning.

| Benchmark | MLLM | Integration of MLLM and Policy | Policy | Task completed in a row (%) ↑ | | | | | Avg. Len ↑ |
| --- | --- | --- | --- | --- | --- | --- | --- | --- | --- |
| | | | | 1 | 2 | 3 | 4 | 5 | |
| CALVIN | Frozen | w CLIP Loss | FT | 94 | 80 | 64 | 51 | 41 | 3.30 |
| | Frozen | w/o CLIP Loss | FT | 90 | 74 | 61 | 54 | 40 | 3.33 |
| | FT | w CLIP Loss | FT | 96 | 83 | 68 | 58 | 48 | 3.53 |
| | FT | w/o CLIP Loss | FT | 88 | 72 | 56 | 46 | 30 | 3.13 |

**Motivation for Further Study.** Although incorporating CLIP loss enables effective performance when fine-tuning the MLLM, this setup may compromise the model's intrinsic generalization ability. This raises a key question: Can we preserve the generalization capability of the MLLM while still achieving effective task-specific adaptation and coordination with the downstream policy?

**Prompt Tuning Strategy.** To solve this problem, we propose a prompt tuning strategy in Dual-system VLA model, illustrated in Figure 3. Specifically, we introduce a learnable <ACT> token into the vocabulary and train only the language modeling head, keeping all other MLLM parameters frozen. This approach enables the model to acquire task-specific knowledge through a lightweight, localized update, without altering the core MLLM parameters—thereby preserving generalization while facilitating improved coordination at the dual-system interface. We validate this method experimentally in Table 3.

**Analysis.** Prompt tuning achieves performance comparable to other training paradigms in the standard CALVIN environment. However, under language generalization evaluation, it substantially outperforms both frozen and fine-tuned setups. Notably, even without CLIP loss supervision, prompt tuning demonstrates improved generalization, suggesting that it minimally interferes with the MLLM's pretrained capabilities. These results demonstrate that prompt tuning is an effective strategy for balancing adaptability and generalization in dual-system VLA models.

Table 3: Further comparison of success rates for different training strategies of the high-level MLLM on the CALVIN and CALVIN-E benchmarks. FT denotes fine-tuning, and PT denotes prompt-tuning.

| Benchmark | MLLM | Integration of MLLM and Policy | Policy | Task completed in a row (%) ↑ | | | | | Avg. Len ↑ |
|---|---|---|---|---|---|---|---|---|---|
| | | | | 1 | 2 | 3 | 4 | 5 | |
| CALVIN | PT | w CLIP Loss | FT | 94 | 78 | 62 | 52 | 42 | 3.28 |
| | PT | w/o CLIP Loss | FT | 94 | 77 | 67 | 60 | 47 | 3.45 |
| CALVIN-E | PT | w CLIP Loss | FT | 81 | 54 | 41 | 27 | 15 | 2.09 |
| | PT | w/o CLIP Loss | FT | 72 | 55 | 40 | 26 | 20 | 2.13 |
| | FT | w CLIP Loss | FT | 76 | 49 | 30 | 15 | 4 | 1.74 |
| | Frozen | w CLIP Loss | FT | 72 | 37 | 21 | 11 | 5 | 1.46 |

## 5.3 STUDY ON POLICY TRAINING STRATEGY (Q3)

In previous approaches, policy training methods can be generally divided into two categories: training from scratch (Zhang et al., 2024; Bu et al., 2024) or fine-tuning a pre-trained model (Shentu et al., 2024). Therefore, understanding the impact of these two paradigms is essential for fair comparison and reproducible research.

Table 4: Success rate of different training strategies for the low-level policy on CALVIN.

| Policy | Task completed in a row (%) ↑ | | | | | Avg. Len ↑ |
|---|---|---|---|---|---|---|
| | 1 | 2 | 3 | 4 | 5 | |
| Fine-tuning | **96** | **83** | **68** | **58** | **48** | **3.53** |
| From-scratch | 89 | 71 | 49 | 42 | 34 | 2.85 |

**Setup.** To ensure a rigorous and fair comparison, we standardize the large model configuration across all experiments by following the LCB structure. The only variable is the downstream policy: one variant uses a pre-trained 3DDA policy, while the other is trained from scratch. All other settings remain fixed.

**Analysis.** As shown in Table 4, policies that are fine-tuned from a pre-trained model consistently outperform those trained from scratch, achieving both higher performance and significantly faster convergence. This demonstrates the clear advantage of leveraging pre-trained policies as initialization, likely because they provide a strong prior and facilitate more efficient learning. Based on these findings, we adopt the **fine-tuning from a pre-trained policy** paradigm as the default configuration for all subsequent experiments, in order to maximize performance and training efficiency.

## 5.4 STUDY ON LATENT TRAINING STRATEGY WITH PROJECTOR PRE-ALIGNMENT (Q4)

Building on the above findings, we observe that pairing a pre-trained policy with a prompt-tuned MLLM yields the best overall performance. However, the connection between the MLLM and the downstream policy still demands careful design due to the potential semantic gap in their representations. To investigate this integration, we conduct an ablation study on the role of the connecting projector.

**Setup.** To bridge the representational gap between the upstream MLLM and the downstream policy, we introduce an MLP projector as an intermediate module. We compare two training strategies: (1) end-to-end training of the entire dual system; and (2) two stage training strategy, where the weight of MLLM is initially frozen while the projector and policy are trained, followed by end-to-end fine-tuning of all components. The difference between these strategies lies in whether the MLP projector is pre-aligned with the downstream policy before end-to-end optimization. Experimental results for these approaches are summarized in Table 5.

**Analysis.** Our results show that without pre-alignment, all types of MLLM configurations fail, regardless of whether they are frozen, fine-tuned, or prompt-tuned. This highlights the critical role of projector in aligning latent spaces and enabling effective communication between modules. Moreover, while a two-stage training process is not necessary when both the projector and the downstream

policy are trained from scratch, Table 4 demonstrates that this alternative yields significantly inferior performance. These findings further emphasize the necessity of pre-training and careful modular coordination to achieve optimal system behavior.

Table 5: Success rate comparison of different projector training strategies for learning latent representations on the CALVIN benchmark. Pre-alignment refers to training the projector prior to training the MLLM. FT denotes fine-tuning, and PT denotes prompt-tuning.

| Benchmark | Pre-alignment | MLLM | Integration of MLLM and Policy | Policy | Task completed in a row (%) ↑ | | | | |
|---|---|---|---|---|---|---|---|---|---|
| | | | | | 1 | 2 | 3 | 4 | 5 |
| CALVIN | ✓ | Frozen | w CLIP Loss | FT | 94 | 80 | 64 | 51 | 41 |
| | ✓ | FT | w CLIP Loss | FT | 96 | 83 | 68 | 58 | 48 |
| | ✓ | PT | w/o CLIP Loss | FT | 94 | 77 | 67 | 60 | 47 |
| | ✗ | Frozen | w CLIP Loss | FT | 0 | 0 | 0 | 0 | 0 |
| | ✗ | FT | w CLIP Loss | FT | 0 | 0 | 0 | 0 | 0 |
| | ✗ | PT | w/o CLIP Loss | FT | 0 | 0 | 0 | 0 | 0 |

## 5.5 STUDY ON TESTING STRATEGY OF THE DUAL SYSTEM (Q5)

A notable advantage of dual-system models is their ability to perform asynchronous inference between System 1 and System 2. To better understand its effect, we conduct extensive experiments analyzing how asynchronous inference influences model performance and task success rate.

**Setup.** To assess the impact of asynchronous inference, we vary the number of low-level action steps executed per high-level MLLM inference step, testing a range from 1 to 60. Our experiments are conducted on the CALVIN-D benchmark. Notably, the maximum action duration for a single command in the 3DDA policy is 60 environment steps.

**Analysis.** As illustrated in Figure 4, the model's performance remains remarkably stable across all tested settings for asynchronous step intervals, even under dynamic environmental conditions. This result is somewhat unexpected, as it suggests that the frequency of high-level updates has limited influence on overall task success—even as the environment evolves. Intuitively, one would expect that more frequent high-level updates would enhance adaptability in dynamic scenarios. These counterintuitive findings imply that the current MLLM exhibits a high degree of insensitivity to changes in the visual scene.

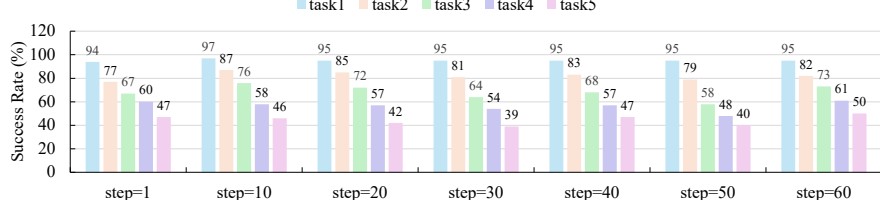

Figure 4: **Evaluations on hierarchical inference.** We evaluate the performance of the dual system on the CALVIN benchmark, with inference steps set to 1 and 60, respectively."Steps" refers to the inference steps of action policy during a single MLLM inference step. The longest environmental steps of the action policy Ke et al. (2024) are 60, which means MLLM only inference once and represents the most typical asynchronous scenarios.

**Extended Setup.** To better understand this phenomenon, we further investigate the semantic content encoded in the latent vectors produced by the MLLM. We project the latent action embeddings into a semantic space and measure their similarity to a predefined vocabulary, aiming to interpret what information the MLLM conveys through these tokens. This analysis is performed in a dynamic environment where, for instance, a blue block suddenly appears on the left.

**Extended Analysis.** As shown in Figure 5, two key observations emerge:

1. *Similarity to spatial words over time.* Despite the robot arm's movements, the latent embedding consistently aligns more with "right" than "left," and other spatial prepositions remain largely constant. This suggests the action token encodes a static semantic representation, with the persistent "right" association likely reflecting its broader usage (e.g., "correct") rather than true spatial grounding.

2. *Top-10 most similar words.* The latent embedding mainly reflects the instruction—target object, spatial relations, and action semantics—with occasional irrelevant words. This indicates limited visual reasoning, as the high-level token conveys task semantics more than dynamic visual context.

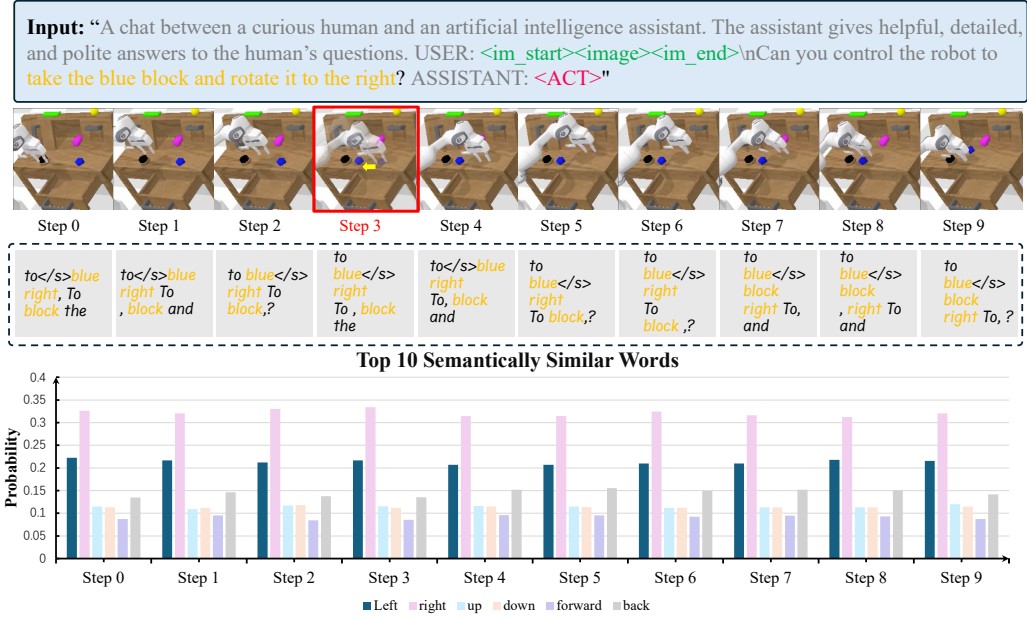

Figure 5: **Evaluation on the shortcoming of existing dual systems.** From top to bottom, the first row displays the input to the MLLM. The second row visualizes a special scenario where, at environment step 3, the blue block is manually shifted to the left. In the third row, we present the top 10 words that are semantically closest to the latent embedding. The bottom row illustrates the probability distribution of spatial words associated with the latent embedding.

## 6 ENHANCING LATENT EMBEDDING LEARNING VIA VISUAL REASONING AUXILIARY TASKS

As revealed in previous experiments, the information currently transmitted through the latent token is insufficient for the downstream model to reliably complete complex tasks. To address this challenge, we introduce a visual reasoning auxiliary task for latent embedding learning. However, most visual reasoning tasks (Shentu et al., 2024) typically require additional annotations, such as sub-task planning. Although VLM/LLM can generate annotations semi-automatically, this process still incurs considerable cost. Thus, we propose a simple and intuitive auxiliary visual reasoning task: by supervising the high-level latent token to predict the target action already included in the dataset, we explicitly encourage the MLLM to attend to both textual and visual information, thereby leveraging its inherent visual reasoning capabilities, as illustrated in Figure 6.

**Setup.** Based on the above conclusions, we employ a two-stage training strategy, where the downstream model is fine-tuned from a pre-trained checkpoint, while the upstream large model adopts a prompt-tuning paradigm. Then, we compare three upstream configurations: a standard MLLM; an LLM-only variant in which visual input is removed from MLLM and the model purely as a large language model; and an auxiliary-task variant, where the latent token is additionally connected to a lightweight action head supervised by an auxiliary loss to predict action. The loss function is defined as follows:

$$\mathcal{L}_{lm}(\texttt{<ACT>}) = \omega_1 \cdot ||\text{MLP}(f_\phi^l(o_t', l')) - a_t^l||$$
$$+ \omega_2 \cdot ||\text{MLP}(f_\phi^r(o_t', l')) - a_t^r||, \quad (1)$$

where $\omega_1$, $\omega_2$ are hyperparameters to balance the effect of each loss item, and MLP represents linear layer. To reconstruct the sequence of 3D locations and 3D rotations, we apply the $L_1$ loss. Details on the architecture and training are provided in Appendix A.3, Appendix A.4, and Figure 6.

**Analysis.** As shown in Table 6, the results show that removing the visual modality and relying solely on the LLM significantly degrades performance, confirming that the MLLM contributes beyond language processing and retains meaningful visual-semantic capabilities. Furthermore, adding auxiliary tasks leads to substantial improvements in success rates. This suggests that the auxiliary losses encourage the MLLM to encode richer visual information into the latent token, thereby enhancing the overall effectiveness of the upstream-downstream interface.

Table 6: Success rate comparison with and without auxiliary tasks for learning latent representation on the CALVIN benchmark. FT denotes fine-tuning, PT denotes prompt-tuning.

| Benchmark | Type of MLLM | Auxiliary Tasks | Policy | Task completed in a row (%) ↑ | | | | | Avg. Len ↑ |
|---|---|---|---|---|---|---|---|---|---|
| | | | | 1 | 2 | 3 | 4 | 5 | |
| | MLLM w/o Vision (PT) | × | FT | 77 | 48 | 26 | 16 | 10 | 1.77 |
| CALVIN | MLLM (PT) | × | FT | 94 | 77 | 67 | 60 | 47 | 3.45 |
| | MLLM (PT) | ✓ | FT | 98 | 92 | 76 | 72 | 63 | 4.01 |

**Comparison to State-Of-The-Art.** We compare the performance of OpenHelix with other state-of-the-art works on CALVIN ABC→D setting. Compared to 3DDA (Ke et al., 2024), building on the above insights, OpenHelix improves the average length on CALVIN ABC→D from 3.83 to 4.08. Moreover, it outperforms the previous dual-system state-of-the-art method, RoboDual (Bu et al., 2024), by 11.5%. Furthermore, although our method appears simple, it outperforms those approaches (Tian et al., 2024; Bu et al., 2025) that rely on complex video generation to guide latent embedding learning, demonstrating the simplicity and efficiency of OpenHelix.

Table 7: Comparison on CALVIN ABC→D. (Environment Step=360)

| Method | Task completed in a row (%) ↑ | | | | | Avg. Len. ↑ |
|---|---|---|---|---|---|---|
| | 1 | 2 | 3 | 4 | 5 | |
| RoboFlamingo (Li et al., 2024) | 82.4 | 61.9 | 46.6 | 33.1 | 23.5 | 2.48 |
| Robodual(Bu et al., 2024) | 94.4 | 82.7 | 72.1 | 62.4 | 54.4 | 3.66 |
| UniVLA(Bu et al., 2025) | 95.5 | 85.8 | 75.4 | 66.9 | 56.5 | 3.80 |
| 3DDA (Ke et al., 2024) | 95.7 | 88.1 | 78.6 | 65.7 | 55.3 | 3.83 |
| Seer(Tian et al., 2024) | 94.4 | 87.2 | 79.9 | 72.2 | **64.3** | 3.98 |
| OpenHelix (Ours) | **97.1** | **91.4** | **82.8** | **72.6** | 64.1 | **4.08** |

## 7 CONCLUSION & LIMITATION

**Conclusion.** Our study highlights the critical role of dual-system vision-language-action (VLA) architectures in achieving robust and adaptable robotic control, particularly in complex, dynamic, and language-guided environments. Through the introduction of the comprehensive DSVLABench suite, we establish a standardized framework for systematic evaluation and comparison of dual-system VLA models. Our experimental results demonstrate that prompt tuning preserves the generalization capabilities of large multimodal language models, while policy fine-tuning from pre-trained models consistently outperforms training from scratch. Notably, we observe that the frequency of high-level updates has limited impact during asynchronous inference, with latent embeddings remaining stable despite dynamic changes. Based on these insights, we propose a simple yet effective dual-system VLA model, OpenHelix. We believe our findings offer the research community valuable guidelines for advancing the development, training, and evaluation of dual-system VLA architectures toward greater generalizability and robustness in robotic applications.

**Limitation.** This work presents the first exploration of a dual-system VLA model, with our current focus primarily on training and evaluation strategies. However, as discussed in Appendix A.2, several important aspects remain to be explored, including the choice of MLLM backbone, downstream policy architecture, and the design of intermediate representations. Additionally, our current experimental results are limited to simulation environments. Further exploration in areas such as real-world robot deployment is a promising direction for future research. Finally, ensuring fast and efficient execution of downstream policies within the dual-system framework remains an open challenge and will be a key focus of our future work.

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

# A APPENDIX

## A.1 A SUMMARY OF DUAL-SYSTEM VLAS.

We introduce recent Dual-System VLA approaches below, with a comparative analysis of their distinctive features summarized in Table 8. **It is important to note that for asynchronous inference to occur, System1 must incorporate real-time perception inputs (such as RGB images).** According to this criterion, approaches like $\pi_0$ (Black et al., 2024), GR00TN1 (Bjorck et al., 2025), and similar methodologies cannot be properly classified within the dual-system framework as they lack this essential characteristic.

**LCB** (Shentu et al., 2024) adopts LLaVA (Liu et al., 2023b) as its System 2. Given a high-level task description and an RGB observation, LLaVA generates a textual action description along with an <ACT> token. The <ACT> token, derived from the final layer, serves as a high-level latent goal. System 1 is a pre-trained 3D Diffusion Actor (Ke et al., 2024) that takes the RGB image, point cloud, and <ACT> token as input to generate actions. System 2 is fine-tuned using LoRA (Hu et al., 2022), while System 1 is fine-tuned in a standard manner.

**DP-VLA** (Han et al., 2024) introduces dual-process theory to justify the rationale behind the dual-system architecture. It presents a more generalizable design choice, where System 2 is not limited to MLLMs (Achiam et al., 2023; Liu et al., 2023b; Radford et al., 2021), but can also be VLA models (Zitkovich et al., 2023; Kim et al., 2024; Team et al., 2024) that are pre-trained on robot data. In experiments, DP-VLA adopts OpenVLA (Kim et al., 2024) as System 2 and uses its encoder to extract latent representations from language instructions and RGB observations to guide System 1. System 1 is implemented using a Transformer architecture (Mandlekar et al., 2022), which encodes RGB images and proprioceptive inputs into actions. System 2 is kept frozen, while System 1 is trained from scratch.

**HiRT** (Zhang et al., 2024) adopts InstructBLIP (Ouyang et al., 2022) as System 2 and utilizes the final-layer representations obtained from encoding both language instructions and RGB observations. These representations are processed with MAP pooling to produce MLLM latent features that guide System 1. System 1 uses an EfficientNet-B3 (Tan & Le, 2019) backbone combined with a MAP block to encode RGB inputs into actions. System 2 is fine-tuned using LoRA (Hu et al., 2022), while System 1 is trained from scratch.

**Robodual** (Bu et al., 2024) adopts OpenVLA (Kim et al., 2024) as System 2 and extracts latent representations from language instructions and RGB observations. It uses both the task latent derived from the instruction and the final action latent as guidance signals. System 1 encodes RGB, depth, tactile, and proprioceptive inputs using a ViT-based encoder (Vaswani et al., 2017), and employs a Perceiver Resampler (Shridhar et al., 2023) to distill key features. A DiT model (Chi et al., 2023) then generates actions by conditioning on the distilled features, the task latent, and a noisy action input. System 2 is fine-tuned using LoRA, while System 1 is trained from scratch.

Table 8: Method comparisons of dual-system VLA models. Here, L, R, P, D, T, and PC represent different modalities: Language, RGB, Proprioception, Depth, Tactile, and Point Cloud, respectively. FT denotes fine-tuning. Pretrain and Scratch denote fine-tuning a pre-trained policy head and training a policy head from scratch, respectively.

| Method | System 2 | | | Latent Rep. | System 1 | | |
|---|---|---|---|---|---|---|---|
| | Model | Input | Training | | Policy Head | Sensory | Training |
| LCB (Shentu et al., 2024) | LLaVA-7B | L+R | Lora FT | Lang(<ACT>) | 3D Diffusion Actor (Ke et al., 2024) | R+P+PC | Pretrain |
| DP-VLA (Han et al., 2024) | OpenVLA-7B | L+R | Frozen | Vis+Lang | Transformer | R+P | Scratch |
| HiRT (Zhang et al., 2024) | InstructBLIP-7B | L+R | Lora FT | MaxPooling(Vis+Lang) | RT-1 (Brohan et al., 2023b) | R | Scratch |
| Robodual (Bu et al., 2024) | OpenVLA-7B | L+R | Lora FT | Action+Lang | DiT | R+D+T+P | Scratch |
| DexVLA (Wen et al., 2025a) | Qwen2-VL-2B | L+R | Lora FT | Lang | ScaledDP (Zhu et al., 2024) | R+P | Scratch |
| Helix | N/A | L+ R + P | N/A | N/A | Transformer | R+P | N/A |

## A.2 KEY DESIGN OF DUAL-SYSTEM VLA MODELS

The central challenge lies in designing the architecture of the two systems and structuring the information flow from the slower, deliberative component (System 2) to the faster, reactive component (System 1) in a manner that preserves the strengths of the former while effectively guiding the latter to execute robotic actions. Achieving this balance is crucial for developing robotic systems that

are both performant and generalizable. As illustrated in Figure 1, addressing this objective requires solving several core design issues:

**1. MLLM Selection.** The requirements for MLLMs vary across different VLA scenarios, and selecting a suitable foundation model is critical for achieving robust performance in robotic tasks. For example, the foundation model used in Flower (Reuss et al., 2025) demonstrates strong spatial awareness and low-level visual capabilities, enabling it to achieve state-of-the-art performance across a range of benchmarks. In contrast, MiniVLA (Belkhale & Sadigh, 2024) adopts Qwen-VL 0.25B (Wang et al., 2024) as its base model to reduce inference costs and computational overhead. These examples illustrate the growing need to identify MLLMs that are both lightweight and sufficiently capable for robotic applications, especially as the landscape of vision-language models evolves rapidly.

Another open question is whether pre-training MLLMs on robotics-specific datasets is necessary. Such pre-training not only narrows the domain gap but also improves the model's robustness in following diverse language instructions, as evidenced by experiments in Robodual (Bu et al., 2024). Determining the trade-off between general-purpose and robotics-specific MLLM pretraining remains an important direction for future work.

**2. Policy Selection.** The choice of small models is relatively less controversial, with the current general consensus being that models based on DiT structure and Flow Matching structure can both meet current needs. However, with the introduction of new policy models such as CARP (Gong et al., 2024), Dense Policy (Su et al., 2025), and other new architectures, downstream small models may also see new designs. Additionally, like Robodual (Bu et al., 2024), whether downstream small models need more modal information, and which modal information is essential for system1, is also a potential question.

**3. Latent Feature Representation Selection.** The selection of latent feature representations is one of the most complex and underexplored aspects of dual-system VLA models. Existing methods vary widely in their design choices, and addressing this gap requires a comprehensive view that includes not only dual-system architectures but also insights from single-system approaches (Black et al., 2024; Bjorck et al., 2025; Li et al., 2024). For instance, DP-VLA (Han et al., 2024) directly uses the final-layer hidden embedding of the MLLM as the latent representation. In contrast, GR00T-N1 (Bjorck et al., 2025) selects hidden states from intermediate layers, motivated by the hypothesis that mid-level features may contain richer visual information and enable more efficient inference. Further, RoboFlamingo (Li et al., 2024) and HiRT (Zhang et al., 2024) aggregate final-layer language and visual features using max pooling as inputs to downstream modules.

Beyond directly extracting hidden states, some methods introduce learnable latent tokens. For example, LCB (Shentu et al., 2024) introduces a dedicated `<ACT>` token that is jointly optimized with the rest of the model to facilitate upstream–downstream communication. This strategy has shown promising results in improving alignment. Robodual (Bu et al., 2024) further extends this idea by incorporating multiple `<ACT>` tokens along with final-layer language features to enrich the latent space. Outside the robotics domain, more sophisticated latent selection strategies have been proposed. MetaQuery (Pan et al., 2025) and LEGO (Lai et al., 2024), for instance, use structured querying or compositional mechanisms to dynamically extract task-relevant features from pretrained backbones. In summary, the choice of latent representations plays a pivotal role in enabling effective dual-system coordination. Identifying representations that are both semantically expressive and compatible with downstream policy learning remains an important direction for future research.

**4. MLLM Training Strategy.** A central question in training MLLMs is whether it is possible to preserve their inherent generalization capabilities while achieving effective integration with downstream tasks. Existing approaches primarily include fully frozen models or end-to-end fine-tuning. However, identifying alternative fine-tuning strategies that better balance generalization and task-specific adaptation remains an important direction for future research.

**5. Policy Training Strategy.** An important consideration in training the policy is how to reduce overall training cost. Leveraging a pre-trained policy and fine-tuning it for the target task can significantly reduce training time and resource consumption. In contrast, training the policy from scratch may lead to longer convergence times and increased instability, particularly due to the potential mismatch in optimization objectives between upstream and downstream components. Whether such objectives hinder convergence remains an open question and warrants further investigation.

**6. Dual-System Integration (Latent Training) Strategy.** A core challenge in dual-system VLA models is how to effectively embed latent information from the high-level MLLM (System 2) as conditioning input into the downstream low-level policy (System 1). In LCB (Shentu et al., 2024), the authors proposed using CLIP loss to constrain the upstream latent features to remain close to the original CLIP text embedding, thereby facilitating the connection between upstream and downstream components. However, this design inherently limits the model's generalization ability, as it forces the latent representations to align with training-specific embeddings and reduces the semantic flexibility of the MLLM. Furthermore, due to inevitable dimensional mismatches between upstream and downstream components when introducing new embeddings, it is common to insert a projector module between them. The training of this projector requires careful design. In particular, when the downstream policy is pre-trained, it is important to first align the projector without updating the MLLM. If both components are unfrozen and trained simultaneously, training becomes unstable and often fails to converge. These observations indicate that the integration strategy, especially the design and training procedure of the projector, plays a critical role in enabling stable and effective communication between the two systems.

**7. Dual-System Asynchronous Strategy.** Lastly, there are asynchronous strategies for dual-system models. LCB (Shentu et al., 2024), HiRT (Zhang et al., 2024), and Robodual (Bu et al., 2024) employ different asynchronous approaches, with LCB (Shentu et al., 2024) being the most naive, using synchronous training but asynchronous testing. Theoretically, differences in inference frequency between upstream and downstream components could affect final performance. However, this is not entirely accurate - if the upstream features being provided aren't effective to begin with, perhaps asynchronous inference between upper and lower layers is merely a pseudo-requirement. Therefore, more experiments are needed to verify this.

In this work, we standardize experimental conditions 1, 2, 3, and 7 to ensure consistency, and focus our evaluation primarily on conditions 4, 5, and 6. These conditions involve widely applicable techniques that are largely independent of the specific choices made in conditions 1, 2, 3, and 7. Through this controlled comparison, we aim to offer insights that may inspire and guide future research in this domain.

## A.3 ARCHITECTURE DETAILS

Our system comprises two main components: a pre-trained MLLM $f_\phi$ and a pre-trained policy $\pi_\theta$, with parameters $\phi$ and $\theta$, respectively. The MLLM includes a text-only large language model and a vision encoder, which projects images into the embedding space of the language model, allowing for a multi-modal understanding of textual and visual inputs. The pre-trained policy consists of a vision encoder and transformer-based diffusion model. Using multiple cross-attention layers, the diffusion model incorporates a lot of conditioning information, such as 3D scene representations, proprioception information, and condition/instruction tokens from the high-level model. In this work, we leverage LLaVA [27] as the high-level MLLM and 3D Diffuser Actor as the low-level pre-trained diffusion policy. Notably, we use a linear layer to replace the 3D Diffuser Actor's text encoder, aligning the dimension of the latent embedding output by the large model with the input dimension of the low-level policy.

## A.4 TRAINING

**Prompt Tuning.** In order to avoid the degradation of MLLM, we introduce one learnable token `<ACT>` $\in \mathbb{R}^d$ at end of language instruction $l$. The new instruction $l'$ is defined as $l' = \{l, \texttt{<ACT>}\}$. During training, all parameters of MLLM are frozen; we only update the embedding of learnable token `<ACT>`.

**Multimodal Reasoning Learning.** As we discussed in section 5.5, we know that these previous methods do not fully utilize MLLM's visual reasoning capability. Specifically, they align the output of the large MLLM model with the output from the text encoder of CLIP. Using purely textual information to supervise the fine-tuning of the MLLM can lead to the degradation of multimodal reasoning capability. Therefore, we designed an auxiliary task to leverage the multimodal reasoning capability of the MLLM fully. This task is very simple and requires no additional data preparation process. The output embedding $z_t^{\texttt{<ACT>}} = f_\phi(o_t', l')$ from the learnable prompt token is passed through linear layers to predict the action trajectories $\tau_t$ and gripper actions $a_t^g$. Through supervised

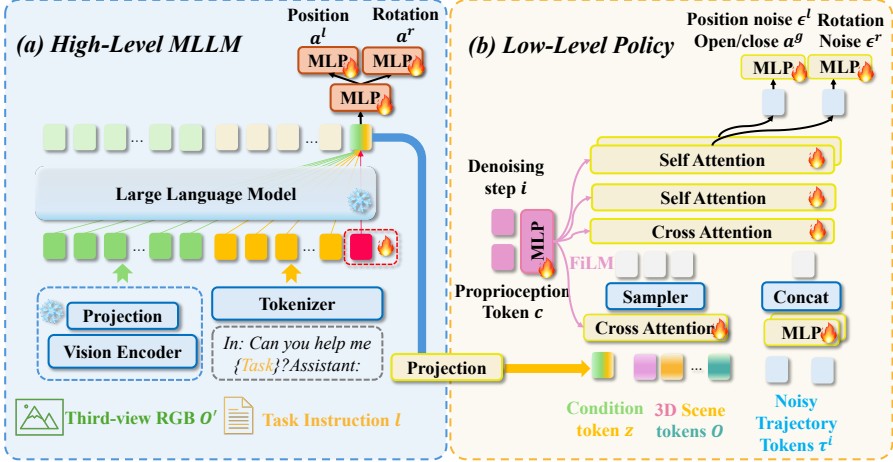

Figure 6: **Detailed framework.** (a) The high-level MLLM (left) takes a third-person RGB image, task instruction, and a learnable token as input. After processing through the large language model, we extract the feature embedding of this token from the final layer as the latent goal for the low-level policy. To fully utilize the MLLM's multimodal reasoning capability, we propose an auxiliary task that uses MLPs to predict the action (including position, rotation, and open/close state) based on this feature embedding, ensuring it captures both visual and textual information. (b) The low-level policy (right) receives the latent goal from the high-level MLLM, combines it with 3D scene tokens and proprioception tokens, and iteratively predicts action noise to generate an accurate action trajectory and gripper state. Notably, our approach keeps all parameters of the MLLM frozen and only fine-tunes the learnable prompt to adjust the MLLM's output, which significantly reduces training costs compared to previous methods.

training on this task, we ensure that the large model has to utilize visual input information and that the latent embedding contains a blend of multimodal information. The loss function is defined as follows:

$$
\begin{aligned}
\mathcal{L}_{lm}(\texttt{<ACT>}) = \ &\omega_1 \cdot ||\mathrm{MLP}(f_\phi^l(o_t', l')) - a_t^l|| \\
&+ \omega_2 \cdot ||\mathrm{MLP}(f_\phi^r(o_t', l')) - a_t^r||,
\end{aligned}
\tag{2}
$$

where $\omega_1, \omega_2$ are hyperparameters to balance the effect of each loss item, and MLP represents linear layer. To reconstruct the sequence of 3D locations and 3D rotations, we apply the $L_1$ loss.

**Diffusion Learning.** Following the previous diffusion-based approach (Chi et al., 2023; Ke et al., 2024; Ze et al., 2024), we train our model using the action denoising objective. During training, we randomly sample a time step $t$ and a diffusion step $i$, adding noise $\epsilon = (\epsilon^l, \epsilon^r)$ to a ground-truth trajectory $\tau_t^0$. The objective is defined as:

$$
\begin{aligned}
\mathcal{L}_{policy}(\theta, \texttt{<ACT>}) = \ &\mathrm{BCE}(\pi_\theta^g(o_t, z_t^{\texttt{<ACT>}}, c_t, \tau_t^i, i), a_{t:t+T}^g) \\
&+ \omega_3 \cdot ||\epsilon_\theta^l(o_t, z_t^{\texttt{<ACT>}}, c_t, \tau_t^i, i) - \epsilon_{t:t+T}^l|| \\
&+ \omega_4 \cdot ||\epsilon_\theta^r(o_t, z_t^{\texttt{<ACT>}}, c_t, \tau_t^i, i) - \epsilon_{t:t+T}^r||,
\end{aligned}
\tag{3}
$$

where $\omega_3, \omega_4$ are also hyperparameters to balance loss items. Please refer to [1] for the details of the loss function.

**Two stage training.** We adopt a two-stage training approach to train our proposed dual system. In the first stage, to initially align the embedding produced by the MLLM with the feature space of the pre-trained policy, we freeze the parameters of the large model and the low-level policy, training only the prompt and projection layers. In the second stage, we keep the large model frozen and unfreeze the low-level policy, fine-tuning it together with the prompt and projection. The objectives in both stages remain unchanged. The only difference between the two stages is whether the low-level policy is frozen. In summary, our loss function includes two components and can be defined as follows:

$$
\mathcal{L}_{total} = \mathcal{L}_{lm} + \mathcal{L}_{policy}
\tag{4}
$$

Table 9: Results on CALVIN ABC-D: We report both success rates and average task completion length (out of 5 tasks) per evaluation sequence. MLLM (PT) denotes our proposed prompt tuning method for MLLM training. Policy(P) indicates loading from a pretrained policy model. Asy(10) represents inference with a 10-step time delay. AUX denotes the additionally introduced auxiliary tasks. EP = 360 indicates that each task is allowed up to 360 environment steps for execution.

| Type | Method | Task completed in a row (%) ↑ | | | | | Avg. Len. ↑ |
|---|---|---|---|---|---|---|---|
| | | 1 | 2 | 3 | 4 | 5 | |
| CALVIN | Only Policy | 92.2 | 78.7 | 63.9 | 51.2 | 41.2 | 3.27 |
| | MLLM (PT) + Policy(P) | 92.2 | 79.2 | 65.0 | 52.9 | 40.9 | 3.30 |
| | MLLM (PT) + AUX + Policy(P) + Asy(10) | **93.3** | **81.8** | **67.9** | 56.6 | 46.0 | **3.45** |
| | MLLM (PT) + AUX + Policy(P) + Asy(60) | 92.8 | 79.7 | 67.5 | **57.3** | **46.9** | 3.44 |
| | MLLM (PT) + AUX + Policy(P) + Asy(60) (EP=360) | 97.1 | 91.4 | 82.8 | **72.6** | **64.1** | 4.08 |
| CALVIN-E | Only Policy | 65.2 | 39.1 | 20.3 | 11.7 | 6.1 | 1.42 |
| | MLLM (PT) + Policy(P) | 71.3 | 44.9 | 28.4 | 17.5 | 10.3 | 1.72 |
| | MLLM (PT) + AUX + Policy(P) + Asy(10) | **78.9** | **57.1** | **40.2** | **29.5** | **20.2** | **2.26** |
| | MLLM (PT) + AUX + Policy(P) + Asy(60) | 78.1 | 56.5 | 38.9 | 27.0 | 19.5 | 2.20 |
| | MLLM (PT) + AUX + Policy(P) + Asy(60) (EP=360) | 84.3 | 66.8 | 49.4 | 35.2 | 24.3 | 2.60 |

## A.5   RESULTS

Based on the results presented in Table 9, the following conclusions—consistent with those from the previous empirical study—can be drawn:

1. **Importance of Prompt Tuning:** The integration of upper and lower layers via prompt tuning has been shown to substantially enhance performance, especially in language generalization scenarios.

2. **Effectiveness of Auxiliary Tasks:** The inclusion of additional auxiliary tasks significantly enhances both standard task performance and generalization capability, primarily by strengthening the model's action proficiency.

3. **Minimal Impact of Asynchronous Inference:** Asynchronous inference has a negligible effect on the inference performance of the general task model. Even when asynchronous inference is performed only once (Asy (60)), the final performance remains largely unchanged.

