# OpenReview forum: "Openhelix: Empirical Analysis of Dual-System VLA Models for Robotic Manipulation"
_ICLR.cc/2026/Conference — Submitted to ICLR 2026_

### Official Review · Reviewer_oK3Z · 2025-10-26

**Soundness:** 3
**Presentation:** 4
**Contribution:** 3
**Rating:** 8
**Confidence:** 4

**Summary:**

The paper “OpenHelix: Empirical Analysis of Dual-System VLA Models for Robotic Manipulation” presents a comprehensive empirical study of dual-system vision-language-action (VLA) architectures, which combine a high-level multimodal large language model (MLLM; System 2) and a low-level reactive policy (System 1). The authors propose a unified evaluation suite called DSVLABench, covering both static (CALVIN-E) and dynamic (CALVIN-D) robotic manipulation benchmarks.
Through controlled experiments, they explore five key questions: (1) the necessity of true dual-system designs, (2) best training strategies for the MLLM, (3) best training strategies for the low-level policy, (4) the importance of latent projector pre-alignment, and (5) the effects of asynchronous inference.
Key findings include that prompt tuning preserves MLLM generalization while improving coordination, fine-tuning pre-trained policies outperforms training from scratch, and pre-aligned projectors are essential for stable latent communication. Interestingly, they observe that asynchronous update frequency between the two systems barely affects performance. Building on these insights, the paper introduces OpenHelix, a simple but effective dual-system model that achieves state-of-the-art results on CALVIN benchmarks.

**Strengths:**

- The paper conducts one of the most thorough empirical studies of dual-system VLA models to date, carefully standardizing training and evaluation conditions. The DSVLABench suite provides a valuable resource for future comparisons and ensures reproducibility.

- The ablation studies convincingly identify what matters in these architectures—showing, for instance, that prompt tuning retains generalization, projector pre-alignment is crucial, and pretrained policies dramatically accelerate convergence. These practical insights are highly useful to the community.

- The extended CALVIN-E (language diversity) and CALVIN-D (dynamic object motion) benchmarks represent meaningful contributions that test both semantic generalization and real-time adaptability—two core challenges in embodied learning.

- Despite using a straightforward architecture, OpenHelix achieves state-of-the-art results, surpassing complex prior models (e.g., Robodual, Seer) on multiple metrics. This validates the paper’s claim that design consistency and modular coordination can outperform architectural complexity.

**Weaknesses:**

- While the empirical findings are robust, the methodological contribution (OpenHelix) mainly aggregates established best practices. There is little theoretical insight into why prompt tuning and asynchronous inference behave as observed, and the “dual-system” framing draws heavily from prior works like DP-VLA and LCB.

- All experiments are conducted in simulation (CALVIN environments). Given the paper’s emphasis on real-time inference and dynamic adaptation, demonstrating transfer to real hardware would have significantly strengthened the claims.

- The discovery that asynchronous inference frequency has minimal impact is intriguing but insufficiently analyzed. It raises questions about whether the high-level embeddings are too static or the MLLM insufficiently sensitive to evolving visual context.

- Some aspects of DSVLABench overlap with prior frameworks like RoboBench or RT-X comparisons. The novelty lies more in consolidation than in conceptual or algorithmic innovation.

**Questions:**

- Could the negligible impact of asynchronous inference suggest that System 2’s latent token is semantically static? Have you measured the variance of latent embeddings across steps?

- How scalable is the proposed setup when using larger backbones (e.g., Qwen2-VL-7B instead of LLaVA-7B)?

- Would integrating explicit feedback from System 1 (e.g., action outcome or success cues) into System 2 improve adaptability in dynamic settings?

- Are there observed limitations when prompt tuning is applied to MLLMs pretrained with different modality alignments (e.g., OpenVLA vs Qwen2-VL)?

---

> ### Author Response · Authors · 2025-11-29
> **Response to the Reviewer oK3Z (part1)**
>
> Thank you very much for your positive comments on our paper.
>
> ##### 4.1 About the deep analysis of proposed method
>
> Your point is very reasonable: in this work we indeed approach the problem from a more **macro / empirical** perspective. While we are not yet able to provide a fully rigorous mathematical theory, we can further clarify the mechanisms behind
>
> ------
>
> ##### 4.1.1Why we use prompt tuning
>
> Regarding the need for **prompt tuning**, this is not our only possible choice. For example, we could also adopt **knowledge isolation** as in the pi-series models. However, at a fundamental level, these approaches are all trying to address the **same core problem**:
>
> > How can we perform VLA tasks **without destroying** the language and semantic capabilities of the underlying model?
>
> This is tightly related to the question of **why** we want to use LLMs for robotics in the first place: we hope to preserve **general scene and language understanding**, rather than training a highly specialized but narrow policy.
>
> However, as is well documented (e.g., in ChatVLA and related work), current VLA systems often undergo **heavy, task-specific fine-tuning**, which tends to **severely damage** the model’s original language and semantic abilities. This forces us to explicitly consider **how to maintain generalization** while adapting to robotics tasks.
>
> From this perspective, both **prompt tuning** and **knowledge isolation** are just two different means toward a **shared objective**: preserving the model’s broad language and scene understanding while enabling effective VLA behavior.
>
> ------
>
> ##### 4.1.2 Why asynchronous inference did not show the expected effects (for now)
>
> Regarding why **asynchronous inference** did not exhibit the strong effects we initially expected, the key reason is the **nature of the information** provided by the upstream model.
>
> Currently, the upstream VLM is **not fine-tuned** for robotic control. As a result, its outputs remain largely at a **semantic / scene-level**:
>
> - It focuses more on **coarse object-level information** (what objects are present, what the scene roughly is),
> - and is relatively **insensitive** to the fine-grained **pose / position** details that are critical for precise manipulation.
>
> In other words, for the upstream VLM in its current form, *where exactly* the object is is less important than *whether* the object is present and what it is. In many of our tasks, as long as the object remains in view, such coarse semantic information is sufficient and **not strongly affected** by moderate asynchrony.
>
> Consequently, even when we perform **synchronous training but asynchronous inference**, the effect on performance is limited under the present setup, because the upstream signal is too coarse to reflect the timing misalignment at a fine spatial level. Our current empirical observations therefore **only support a conclusion at this stage** of model capability.
>
> We also want to stress that this is **not a universal or permanent claim**. If, in the future, we have a VLM that is **highly sensitive to fine-grained pose and spatial relations**—that is, one whose outputs change significantly with small variations in object position—then our conclusions about asynchronous inference may no longer hold. In that regime, inference latency and synchronization would likely have a much more pronounced impact.
>
> In summary, our current findings about asynchronous inference should be interpreted as **contingent on the present generation of VLMs**, which still operate primarily at a coarse semantic level, rather than as a definitive statement about all future dual-system designs.

---

> ### Author Response · Authors · 2025-11-29
> **Response to the Reviewer oK3Z (part2)**
>
> ##### 4.2 About the real-world experiments
>
> Regarding real-robot experiments, due to time constraints we only conducted three types of tasks: **Lift, Push, and Stack**.
>
> Under standard settings, these tasks achieve very high success rates. Therefore, to specifically evaluate **semantic robustness**, we modified the language instructions. Concretely, the original commands:
>
> - “Lift the watermelon”
> - “Put the pepper in the bowl”
> - “Stack green bowl on yellow bowl”
>
> were replaced with more complex, **previously unseen** descriptions, such as:
>
> - “Grasp the green, striped spherical object.”
>
> The results are as follows. These experiments demonstrate that the **OpenHelix** architecture preserves strong **semantic generalization**, rather than overfitting to specific templated commands. This observation is consistent with our simulation findings and further confirms the **generalizability** of our method.
>
> | **Model** | **Lift** | **Put** | **Stack** | **Average** |
> | :-------- | :------- | :------ | :-------- | :---------- |
> | 3DDA      | 25       | 20      | 5         | 16.7        |
> | **Ours**  | **90**   | **74**  | **92**    | **85.3**    |
>
> ##### 4.3  About asynchronous inference
>
> Thank you for bringing up this point; we clarify it in more detail below.
>
> For **asynchronous inference**, similar to LCB, our **high-level embeddings** are primarily intended to carry **semantic information**, since the downstream policy already has access to **real-time perceptual input**. In this setting, having language instructions with strong generalization capability is particularly valuable.
>
> However, as shown in **Figure 5**, because the MLLM is **frozen**, the information it encodes is largely determined by its **pretraining on generic vision–language tasks**. As a result, the high-level embeddings are predominantly **texture- and object-centric**, which is consistent with the fact that our training corpus contains a large number of **grounding-style examples**.
>
> Precisely due to being frozen, the MLLM is **not very sensitive** to the finer-grained **pose and positional details** that are crucial for robotic manipulation. Instead, it mainly focuses on **coarse, object-level properties**: the exact location of an object is less important, as long as the object is present. This is usually sufficient for generic VL benchmarks, but not for precise control. From this perspective, the statement that the **high-level embeddings are “too static”** is reasonable.
>
> This motivates our proposal to introduce **auxiliary tasks** that *selectively enhance* this aspect of the representation. Conceptually, this is very similar to:
>
> - the **pre-finetuning of the VLM** adopted in pi0.5, and
> - the grounding-based finetuning strategy used in **GROOT**.
>
> These approaches are essentially **equivalent in paradigm**: they act as a form of **regularization**, analogous to **REPA**, where one imposes appropriate constraints on certain **intermediate representations** to improve both **downstream adaptability** and **fast adaptation**.
>
> Under this view, the conclusion that the current **MLLM is insufficiently sensitive to evolving visual context** is also natural: its semantics are robust but relatively static with respect to fine-grained spatial changes. Our auxiliary-task design is precisely aimed at addressing this limitation within the same dual-system framework.
>
> #####

---

> ### Author Response · Authors · 2025-11-29
> **Response to the Reviewer oK3Z (part3)**
>
> ##### 4.4  About DSVLABench
>
> We agree with your assessment that **DSVLABench itself is not a major algorithmic innovation**. Its main contribution lies in providing a benchmark that explicitly targets the two core capabilities of Dual-System architectures—**generalization** and **dynamic-scene robustness**—and in offering a more systematic way to evaluate them.
>
> At the same time, we must also acknowledge a practical constraint: as we emphasized in the paper, there is currently **no unified standard** in the Dual-System literature. To remain consistent with prior work, we had to adopt the **metrics and experimental setups that most existing studies report**. In addition, because the majority of Dual-System methods are **not open-sourced**, **CALVIN** is essentially the only benchmark that enables a reasonably fair comparison at this stage. We were aware of this limitation, which is why we designed **multiple variants on CALVIN** to more comprehensively capture and contrast the behaviors of different models.
>
>
>
>
> ##### 4.5  About variance of latent embeddings across steps
>
> Regarding this question, I agree with your perspective. Although we have not performed exactly the experiment you suggested, we believe a related experiment we conducted offers supporting evidence for your intuition.
>
> Specifically, we took a real-world **first-person video** in which the camera walks toward **three balloons of different colors**, eventually stopping directly in front of one balloon. From this video, we extracted all frames and, for each frame, computed the corresponding **latent representation** using the same VLM and the same textual query.
>
> We then measured the **cosine similarity** between the latents of **adjacent frames**. The results show that cosine similarity changes **sharply only when the number of visible balloons decreases** (i.e., as the camera approaches, some balloons move out of the field of view). At all other times, when the same balloons remain visible and only their **distance / relative position** changes, the cosine similarity remains almost unchanged.
>
> This suggests that the current VLM is indeed highly sensitive to **scene-level semantic changes** (e.g., “one balloon disappears from view”), but largely **insensitive to fine-grained positional changes** when the set of visible objects remains the same. In other words, it reacts strongly to “what is in the scene” but much less to “where exactly it is.”
>
> From this perspective, it is reasonable to say that **System 2’s latent tokens** in our setup are currently good at capturing **global scene-level details**, but much less effective at capturing **precise, position-level details**.
>
>
>
> ##### 4.6  About larger model
>
> Returning to the core of your question, our primary goal at this stage is to enable **fair comparisons across different methods**. For this reason, we deliberately keep both the **MLLM choice** and the **downstream policy** consistent with LCB. This allows us to control key variables and isolate the effect of **training strategies** on Dual-System performance, rather than conflating architectural and optimization differences.
>
> We fully agree that using **stronger backbones** is valuable and worth exploring. However, due to computational constraints, we are currently unable to run such experiments in the short term (each full experiment takes about **10 days on 8×A100** GPUs). Instead, we would point readers to concurrent work that adopts a **similar coupling scheme** to ours. For example, **X-VLA** also uses a **prompt-tuning–based interface** (albeit in a single-stage model), and its results across multiple benchmarks clearly demonstrate the **data-scaling capability** of this paradigm.
>
> As for **model scaling**, we believe that, in the context of embodied AI, it is still an **open question** whether larger backbones are always necessary or beneficial. A growing body of work suggests that many current VLA systems are still in a heavily **overfitted regime**, and their true scalability properties remain unclear. In this sense, understanding **if and how** scaling interacts with Dual-System design is itself an important direction for future research, rather than something we can conclusively resolve within the scope of this paper.

---

> ### Author Response · Authors · 2025-11-29
> **Response to the Reviewer oK3Z (part4)**
>
> ##### 4.7  About adaptability in dynamic settings?
>
> We address your concern by combining it with our response to Reviewer B (Bsn4), specifically the part titled *“2.1. As to Section 5.1”*.
>
> The fundamental reason for failure on **dynamic tasks** is that, during training and in the way data are organized, there is an implicit **“static object” assumption**. Once the object is allowed to move, this assumption breaks down. Due to the **inference latency** of large models, by the time the model has produced an action, the object may already have moved a non-negligible distance. In this regime, no matter how we optimize or enlarge the **trunk** (i.e., the action chunk length or planning horizon), the resulting action is still computed from an **inaccurate, outdated observation**. This is a structural limitation that cannot be fixed simply by increasing the trunk size.
>
> Put differently:
>
> - Enlarging the trunk can alleviate inference issues **under the stationary-object assumption**, where the environment remains largely unchanged between observation and execution.
> - However, once objects are moving, **any** finite trunk size will still suffer from **observation–action asynchrony**: the state used for inference is no longer the state at execution time.
>
> Therefore, to address the problem **at its root**, the downstream policy must be given **real-time perceptual input**, and the **control frequency** should be dominated by a **small, fast controller** rather than a large, slow model—or alternatively, one should adopt a **GEN-0–style** approach with tightly coupled perception and control.
>
> From this perspective, simply “adding real-time feedback” on top of a pseudo dual-system does **not** fundamentally solve the issue. A root-level solution should follow the GEN-0 direction: designing architectures where high-frequency, closed-loop control is intrinsic, rather than trying to compensate for large-model latency via longer trunks.
>
> ##### 4.8  About observed limitations when prompt tuning ?
>
> This question is indeed quite open, and we do not yet have a complete answer. However, we can form some preliminary hypotheses by looking at **VLM + T2V**–style tasks.
>
> For example, in the **Meta-Query** work, we observe that the **number of prompts** has a significant impact on performance. Intuitively, prompts are effectively **re-organizing the information** provided by the VLM so that it becomes more suitable for the downstream task. If we continue along this direction, it is likely that we will eventually encounter a BLIP-like challenge:
>
> > What happens when important information is lost in this re-organization or compression?
>
> How to deal with such **information loss** remains an open question. In the long run, it may be more desirable to design **better direct mappings** between VLM representations and downstream modules, rather than relying on an additional intermediate prompt-based layer as the sole interface. This is, in our view, a deeper and more fundamental problem that deserves long-term investigation.
>
> That said, **at the current stage**, prompt-based interfaces are still a highly **efficient and practical** solution, and thus remain a reasonable design choice in our work.
>
> [1] ChatVLA: Unified Multimodal Understanding and Robot Control with Vision-Language-Action Model
> [2] GR00T N1: An Open Foundation Model for Generalist Humanoid Robots
> [3] X-VLA: Soft-prompted transformer as scalable cross-embodiment vision-language-action model
> [4] GEN-0: Embodied Foundation Models That Scale with Physical Interaction
> [5] Transfer between Modalities with MetaQueries

---

### Official Review · Reviewer_9bia · 2025-10-31

**Soundness:** 3
**Presentation:** 3
**Contribution:** 3
**Rating:** 4
**Confidence:** 2

**Summary:**

This paper presents a novel dual-system VLA architecture and introduces a benchmark called DSVLABench, designed to systematically evaluate and compare different VLA architectures. The authors experimentally validate the effective collaboration between the high-level language model and the low-level control policy, and propose an innovative training method (prompt tuning), to enhance the model's generalization ability. Experimental results demonstrate that the proposed OpenHelix model performs excellently in multimodal reasoning tasks, showing superior performance compared to other models.

**Strengths:**

- The authors introduce the DSVLABench benchmark, providing a standardized platform for the comparison and evaluation of VLA architectures, filling the gap in the evaluation framework within this field. They also introduce the strategy of prompt tuning, which effectively enhances the generalization ability of the high-level language model, particularly in cross-modal tasks involving vision and language.
- Despite its simple structure, the proposed OpenHelix model has been systematically evaluated and compared with other dual-system VLA architectures on the CALVIN benchmark. Experimental results demonstrate its outstanding performance in task success rate, highlighting its efficiency.

**Weaknesses:**

- Although the paper shows good results in simulation environments, it lacks validation of the model’s application in the real world, which affects its generalizability and practicality in real-world scenarios.
- While the paper focuses on the integration of the high-level language model and low-level control strategy, as well as the performance of the high-level language model, it provides limited in-depth discussion on the low-level control strategy, especially regarding its performance and optimization in different application scenarios, which requires further exploration.
- Although the paper compares the performance of different models, it lacks a detailed analysis of how the selection of training data and optimization strategies specifically impact the final results.

**Questions:**

- In real-world scenarios, how adaptable and flexible are the low-level control strategies? In the real and complex real world, does the model have sufficient generalization ability?
- The experiments in the paper mainly focus on comparative tests in the simulation environment, but lack comprehensive evaluations under different datasets and task scenarios. Could more abundant experimental Settings be provided further to test the generalization ability and robustness of the model?

---

> ### Author Response · Authors · 2025-11-29
> **3. Response to the Reviewer 9bia (part1)**
>
> ##### 3.1. About  real-world validation
>
> Regarding real-robot experiments, due to time constraints we only conducted three types of tasks: **Lift, Push, and Stack**.
>
> Under standard settings, these tasks achieve very high success rates. Therefore, to specifically evaluate **semantic robustness**, we modified the language instructions. Concretely, the original commands:
>
> - “Lift the watermelon”
> - “Put the pepper in the bowl”
> - “Stack green bowl on yellow bowl”
>
> were replaced with more complex, **previously unseen** descriptions, such as:
>
> - “Grasp the green, striped spherical object.”
>
> The results are as follows. These experiments demonstrate that the **OpenHelix** architecture preserves strong **semantic generalization**, rather than overfitting to specific templated commands. This observation is consistent with our simulation findings and further confirms the **generalizability** of our method.
>
> | **Model** | **Lift** | **Put** | **Stack** | **Average** |
> | :-------- | :------- | :------ | :-------- | :---------- |
> | 3DDA      | 25       | 20      | 5         | 16.7        |
> | **Ours**  | **90**   | **74**  | **92**    | **85.3**    |
>
> ##### 3.2. About  low-level control strategy
>
> While the points you raised are indeed important, we would like to reiterate what we stated in Lines 832–836 of the paper. At this stage, our primary focus is on **how to design effective training strategies for dual-system architectures**, in order to improve performance in **dynamic scenarios** and **language generalization**. To isolate these aspects, we deliberately keep both the choice of **MLLM** and the **low-level controller** fixed across experiments.
>
> For this reason, and to enable a direct comparison with the **canonical dual-system paradigm LCB**, we use **3DDA** as the *only* downstream policy in our study. This allows us to concentrate on the impact of the dual-system design and training scheme, rather than confounding the results with variations in policy architectures.
>
> Due to time constraints, we replaced the downstream policy with a custom **Flow Matching–based policy architecture**, using only **2D inputs** (referred to as **2D FM**). The results are as follows: the performance is largely comparable to our original setup, which demonstrates the **generality and robustness** of our proposed framework.
>
> | Method                     | 1    | 2    | 3    | 4    | 5    | Avg. Len. ↑ |
> | :------------------------- | :--- | :--- | :--- | :--- | :--- | :---------- |
> | LLAVA 7B+ 2D FM            | 96.6 | 90.8 | 83.1 | 75.6 | 67.7 | 4.14        |
> | LLAVA 7B+ 3DDA (Raw Paper) | 97.1 | 91.4 | 82.8 | 72.6 | 64.1 | 4.08        |
>
> We agree that exploring a broader set of downstream policies is valuable, and we plan to investigate this in future work. We sincerely appreciate your suggestion and view it as an important direction for further improvement.
>
> ##### 3.3. About  the selection of training data and optimization strategies
>
> Regarding the **training data**, we use **CALVIN** in order to remain consistent with prior work. Given the current literature, this is effectively the only viable choice if we want to make fair comparisons with existing dual-system approaches.
>
> As for the **optimization strategies**, it is of course possible to obtain higher performance by employing more sophisticated training tricks. However, this is not the primary objective of our paper. Our goal is to study dual-system design under **comparable optimization settings**, and to isolate how architectural and training-scheme choices affect the behavior of dual systems. Introducing too many additional optimization heuristics would add confounding factors and make it difficult to draw clear conclusions about what actually matters.
>
> In this early and somewhat chaotic stage of the field, we believe it is important to **control for such variables** and avoid over-emphasizing aggressive optimization tricks. For readers who are primarily interested in pushing raw performance, we would instead refer to **X-VLA[1]**, which adopts a conceptually similar approach while incorporating stronger optimization and achieves better overall results.

---

> ### Author Response · Authors · 2025-11-29
> **3. Response to the Reviewer 9bia (part2)**
>
> ##### 3.4 Q1 how adaptable and flexible are the low-level control strategies
>
> Regarding your question about the **downstream policy**, our initial choice to use **3DDA** as the only downstream controller was deliberate: it allows us to make a direct and fair comparison with the **canonical dual-system paradigm LCB**. For this reason, we restricted ourselves to 3DDA at the beginning.
>
> Due to time constraints, we replaced the downstream policy with a custom **Flow Matching–based policy architecture**, using only **2D inputs** (referred to as **2D FM**). The results are as follows: the performance is largely comparable to our original setup, which demonstrates the **generality and robustness** of our proposed framework.
>
> | Method                     | 1    | 2    | 3    | 4    | 5    | Avg. Len. ↑ |
> | :------------------------- | :--- | :--- | :--- | :--- | :--- | :---------- |
> | LLAVA 7B+ 2D FM            | 96.6 | 90.8 | 83.1 | 75.6 | 67.7 | 4.14        |
> | LLAVA 7B+ 3DDA (Raw Paper) | 97.1 | 91.4 | 82.8 | 72.6 | 64.1 | 4.08        |
>
> Moreover, additional works such as **VLA-Adapter** and **X-VLA** already offer further evidence in this direction. They adopt **different downstream policy designs** while following a **conceptually similar framework**, which demonstrates that this overall scheme is **inherently extensible** to a wide range of low-level controllers.
>
> ##### 3.5 Q2 More comprehensive experiments.
>
> As we emphasized in the paper, there is currently **no unified standard** in the Dual-System literature. To remain consistent with prior work, we had to adopt the **metrics and setups that most existing studies report**. In addition, because the majority of Dual-System methods are **not open-sourced**, **CALVIN** is essentially the only benchmark that enables a reasonably fair comparison at this stage. We were aware of this limitation, which is why we conducted **multiple variants on CALVIN** to more comprehensively reflect performance differences across models.
>
> For readers who are more interested in the **long-term potential and scalability** of this line of work, **X-VLA** is indeed a valuable reference. It also employs prompt tuning, but is evaluated on a much broader set of tasks and environments, providing a more extensive assessment of scalability.
>
> To further validate the **reasonableness and generality** of our approach beyond CALVIN, we also conducted experiments on **LIBERO-Long**, where we observed consistent trends. This supports that our dual-system design is not overfitted to a single benchmark.
>
> |              | Pi0  | UniVLA | Seer | Ours |
> | :----------- | :--- | ------ | ---- | :--- |
> | Success Rate | 85.2 | 92.0   | 87.7 | 93.8 |
>
> However, in this paper our primary goal is to **minimize confounding variables** and ensure a **clean, controlled comparison** of Dual-System designs. Introducing additional sources of uncertainty (e.g., heterogeneous datasets, backbones, or training tricks) would make it harder to draw clear conclusions about the architectural and training choices we study. We hope this rationale is understandable.
>
> [1] X-VLA: Soft-prompted transformer as scalable cross-embodiment vision-language-action model
> [2] 3D Diffuser Actor: Policy Diffusion with 3D Scene Representations
> [3] LCB: Latent Codes as Bridges in Hierarchical Robot Control
> [4] VLA-Adapter: An Effective Paradigm for Tiny-Scale Vision-Language-Action Models

---

### Official Review · Reviewer_Bsn4 · 2025-11-01

**Soundness:** 3
**Presentation:** 3
**Contribution:** 2
**Rating:** 4
**Confidence:** 4

**Summary:**

The paper proposes an empirical study of various design decisions for dual-system vision-language-action models, given the recent rise in their popularity and large variation in their designs among latest works (which make it difficult to perform controlled comparisons between them). The paper introduces two new benchmarks that are built on top of the existing CALVIN simulation benchmark: CALVIN-E (which assesses high-level semantic understanding, by testing generalization to new language commands) and CALVIN-D (which assesses coordination between the two systems in dual-system VLAs and reactivity to dynamic scenarios). These benchmarks are included in the proposed DSVLABench, which examines five key aspects of dual-system VLAs and provides a systematic comparison of different design decisions. The results of the empirical study lead to five insights related to effects of various components of dual-system VLAs, such as the dual-system architecture, the usefulness of auxiliary training tasks, and the frequency of high-level updates during asynchronous dual-system inference.

**Strengths:**

* The paper conducts a large quantity of experiments, which lead to useful insights about which techniques are necessary or helpful when training dual-system VLAs. The discussions in Section 5.4 and Section 5.5 are particularly interesting, as they reveal that the projector training strategy is crucial when training the dual-system VLA on CALVIN, and that the policy is robust to infrequent high-level updates from the MLLM during evaluations in dynamic tasks.
* The paper introduces two new evaluation suites, CALVIN-D and CALVIN-E, which test robustness to dynamic changes in object position and new language instructions.
* The paper systematically studies individual components in the design of a dual-system VLA policy with various ablations to highlight their effects on policy performance.
* The paper is generally well written and easy to follow.

**Weaknesses:**

* Section 5.1 seems to be missing important details. For example, there is no discussion on "3DDA Ke et al. (2024)" in Table 1, and it is unclear to the reader how this method is evaluated and why it performs much better than the "pseudo dual-system" VLA RoboFlamingo. Elaboration on this comparison would greatly aid the reader. Also, it is not clear to me why pseudo dual-system VLAs necessarily fail in dynamic tasks like the ones used in the proposed CALVIN-D test suite. What happens if the policies are queried much more frequently and action chunks are only partially executed so that the policies are more reactive? Relatedly, what are the querying frequency and action chunk sizes for the two methods tested in Table 1? Further, while the paper comments on the lack of fair comparisons due to the variation in designs of dual-system VLAs, the comparison in Table 1 does not seem like a controlled comparison either, as entirely different architectures and algorithms are used.
* In Section 5.2, while the proposed techniques such as CLIP loss and prompt tuning individually show improvements over other variants, this comparison is conducted with one particular type of MLLM architecture in one particular benchmark, and I have concerns about whether the findings generalize broadly. Therefore, the authors' conclusion that "prompt tuning is an effective strategy for balancing adaptability and generalization in dual-system VLA models" seems too broad and should be qualified. In my opinion, a general claim like this should be supported by multiple pieces of evidence, such as similar findings across more than one architecture/model and more than one task suite. This is especially the case because it is not obvious that, e.g., freezing the MLLM backbone and just doing prompt tuning is a technique that will scale to other task suites (for example, what if this strategy shows limited capacity for fine-tuning at larger dataset scales?). Also, why does prompt tuning lead to worse performance on the base CALVIN suite (3.45) than fine-tuning with CLIP loss (3.53)?
* Fine-tuning outperforming training from scratch in Section 5.3 is not a novel finding in VLA works and is a fairly established understanding at this point. For example, the OpenVLA paper by Kim et al. (CoRL 2024) discusses this already.
* "OpenHelix" is not presented in the main text until Section 6, and it is not clear how it is defined. (It does appear in Figure 1 but without a concrete definition). It would be good to explain what exactly this name refers to, and present it earlier on in the paper.
* Several state-of-the-art prior works are omitted in the CALVIN ABC->D comparisons, including UniVLA (Bu et al., RSS 2025) which gets an average length of 4.41 and Seer-Large (Tian et al., ICLR 2025) which gets 4.28. Why these are not included in the comparisons warrants discussion.
* All of the comparisons are done in the CALVIN simulation benchmark, which adds uncertainty as to whether the findings hold across other benchmarks/tasks. Similar findings in other benchmarks/tasks would strengthen the paper.

Minor:
* The usage of `\citet{}` and `\citep{}` is not correct throughout the paper and the lack of punctuation around citations is distracting to the reader. For example, in the first sentence of the introduction, there are missing parentheses around the cited vision-language-action model works.
* "Roboflmanigo" typo appears multiple times in the paper.

**Questions:**

See questions in the Weaknesses section above.

---

> ### Author Response · Authors · 2025-11-29
> **Response to the Reviewer Bsn4 (part 1)**
>
> Thank you very much for your feedback. Your points regarding the details of the paper are very valid, and we will respond to each of your questions in turn.
>
> ##### 2.1. Regarding the Section 5.1
>
> ##### 2.1.1 Comparison between 3DDA and RoboFlamingo
>
> Regarding **RoboFlamingo**, there is a specific design choice that is crucial for understanding its failure mode in our setting. At inference time, the policy explicitly depends on **model-internal temporal context**: it takes the **latent features from several past frames** as input to an LSTM. This design choice is exactly what causes the model to fail completely in our scenario.
>
> During RoboFlamingo’s training regime, in the phase before the object is grasped, the object is effectively **static**. As a result, the latent features extracted from multiple consecutive frames are almost identical with respect to the object; the model never experiences significant temporal variation in object-related features in this “pre-grasp” phase. In contrast, in the **CALVIN-D** setting, the object is **moving**. Consequently, the latent features across multiple frames differ substantially in the visual stream. This mismatch between the training-time assumption (stationary object, nearly constant latents) and the test-time condition (moving object, varying latents) directly leads to RoboFlamingo’s failure. A related explanation is also provided in Lines 203–209 of the paper, which you may cross-check.
>
> This highlights an important point: many existing models are trained under **implicit, unspoken assumptions**, such as “the target object does not move before grasp.” Once this assumption is violated at test time, the model can fail catastrophically—especially given that most current systems are still in an **overfitting-heavy regime**, with limited robustness beyond their training distribution. This issue is particularly severe for architectures that **explicitly rely on processing historical information** or **asynchronous observations**, since any deviation from the assumed temporal pattern can corrupt the internal state (e.g., LSTM hidden states).
>
> Similar hidden assumptions also include:
>
> - the **camera viewpoint** remains fixed;
> - the **robot base** does not move;
> - the scene layout remains quasi-static.
>
> All of these assumptions implicitly constrain the capability frontier of current VLA models and limit their ability to generalize to more realistic, dynamic scenarios.
>
> By contrast, the **3DDA small model** does not suffer from this particular problem to the same extent. Because the model is relatively **small**, it can update its observations at a **higher frequency** and does **not rely on long histories or asynchronous temporal aggregation**. It primarily reacts to the most recent observation instead of maintaining a fragile dependence on past latent sequences. As a result, its performance drop in the CALVIN-D setting is comparatively smaller.

---

> ### Author Response · Authors · 2025-11-29
> **Response to the Reviewer Bsn4 (part 2)**
>
> ##### 2.1.2 why pseudo dual-system VLAs necessarily fail in dynamic tasks
>
> As also discussed in Sec. 2.1.1, the fundamental reason for failure on **dynamic tasks** lies in a strong **implicit assumption** baked into both the model training process and the way data are organized: namely, that the manipulated object remains **stationary**. Once the object is allowed to move, this assumption breaks down.
>
> For **pseudo dual-system VLAs**, the problem is exacerbated by **inference latency**. Because of the non-negligible forward-pass time, by the time the model has finished computing an action, the object may already have moved a significant distance. In this regime, no matter how we optimize or enlarge the **trunk** (i.e., the chunk size or planning horizon), the resulting action is still computed from an **outdated observation**. This is a structural limitation that cannot be eliminated simply by increasing the trunk length.
>
> Put differently:
>
> - Increasing the trunk size can help under the **stationary-object assumption**, where the environment does not change significantly between observation and execution.
> - However, once the object is moving, **any** finite trunk size will still suffer from **observation–action asynchrony**: the state used for inference is no longer the state at execution time.
>
> Therefore, to address the issue **at its root**, the downstream policy must be given **real-time perceptual input**, and the **control frequency** should be dominated by a **small, fast controller**, rather than by a large, slow model. Alternatively, one can adopt a GEN-0-style approach, where perception and control are tightly coupled in a high-frequency closed loop.
>
> Related to your suggestion that “policies are queried much more frequently and action chunks are only partially executed so that the policies are more reactive”: this setup still inherently suffers from **model inference latency**. For a robot arm, it is sometimes acceptable to let the arm pause and wait for the next decision. But in a **dynamic environment**, the objects will continue to move regardless of whether the robot is waiting. At present, for essentially all VLAs, a single forward pass typically takes on the order of **0.1 seconds**. In an application such as a conveyor belt moving at **5 m/s**, the object will have traveled about **0.5 meters** during one inference step. Yet the model’s action is still conditioned on the **pre-movement observation**, which is already obsolete by the time the action is executed.
>
> For these reasons, **pseudo dual-system VLAs** are fundamentally problematic in truly dynamic, real-world scenarios. Their limitations are simply not exposed yet because current evaluation benchmarks rarely stress this dimension of closed-loop, high-speed interaction.
>
> ##### 2.1.3 more details of 3DDA and RoboFlamingo
>
> We believe that the explanations in Sections 2.1.1 and 2.1.2 already address your concern. In summary, **action chunking** can indeed alleviate inference-speed issues for object manipulation in **static environments**, but it cannot fundamentally solve the problem in **dynamic settings**, because the limiting factor there is the **per-step inference time** of the large model itself.
>
> We also agree that the particular experiment you pointed out is not a perfectly controlled comparison. However, the qualitative behavior it reveals is still meaningful and does support several of our conclusions about the limitations of current architectures. We are planning to add more experimental results to further substantiate these findings.
>
> That said, the computational cost is extremely high: a single ablation experiment currently requires 8 A100 GPUs running for about 10 days. We kindly ask for your understanding regarding the present experimental scope. This is an ongoing effort, and we will include more results as they become available.

---

> ### Author Response · Authors · 2025-11-29
> **Response to the Reviewer Bsn4 (part 3)**
>
> ##### 2.2 Regarding Section 5.2
>
> ##### 2.2.1 whether the findings generalize broadly.
>
> We would like to clarify this point more explicitly.
>
> Our primary reliance on the **CALVIN** benchmark is largely dictated by the current landscape of **mainstream dual-system work**. Most prominent dual-system approaches (e.g., **LCB**, **RoboDual**) conduct their experiments on CALVIN, and the majority of these methods are **not open-sourced**. Consequently, at this stage we are effectively constrained to perform experimental comparisons on this benchmark. While this is not ideal, it is, in practice, the only feasible option for a fair and consistent comparison with existing work.
>
> To further validate the **reasonableness and generality** of our approach beyond CALVIN, we also conducted experiments on **LIBERO-Long**, where we observed consistent trends. This supports that our dual-system design is not overfitted to a single benchmark.
>
> |              | Pi0  | UniVLA | Seer | Ours |
> | :----------- | :--- | ------ | ---- | :--- |
> | Success Rate | 85.2 | 92.0   | 87.7 | 93.8 |
>
> Moreover, to the best of our knowledge, recent methods such as **VLA-Adapter** and **X-VLA** have already adopted similar ideas and extended them to **more realistic environments and a broader set of benchmarks**, which alleviates the concern that this line of research is tied exclusively to CALVIN-like settings.
>
> In particular, **X-VLA** follows a strategy that is closely aligned with ours and reports both **pre-training** and **real-robot** evaluations on a comprehensive set of benchmarks. Their results provide complementary evidence that such architectures can indeed scale beyond CALVIN and directly address the type of concern you raised.
>
> ##### 2.2.2 About prompt tuning lead to worse performance on the base CALVIN suite (3.45) than fine-tuning with CLIP loss (3.53)?
>
> First, we consider the scores 3.45 and 3.53 to be essentially equivalent; such a small numerical difference is practically negligible.**CALVIN-E**, where the gap increases to **2.13 vs. 1.74**.
>
> This is what truly reflects the underlying problem: **fine-tuning with a CLIP loss still substantially degrades the model’s semantic representations**. The performance discrepancy on the more challenging, out-of-domain setting (CALVIN-E), rather than the near-identical in-domain results, is what reveals the damage to semantics caused by this fine-tuning scheme.
>
> ##### 2.3 As to “Fine-tuning outperforming training from scratch ”
>
> These two situations are in fact fundamentally different.
>
> In our work, we are evaluating the **downstream small model** within a dual-system architecture. This model itself is an **independent VLA-capable policy**, which is a different setting from what OpenVLA is facing. The comparison should therefore not be interpreted as if they were the same problem.
>
> Moreover, our conclusion here goes one step further: it sheds light on **how to better handle the training of the downstream small policy** and how to design the **mapping between the upstream and downstream models**. This is precisely about understanding and improving the interface and coordination between the two systems.
>
> For these reasons, we believe this issue needs to be reconsidered from a **different perspective**, rather than being conflated with the OpenVLA scenario.
>
> ##### 2.4 About OpenHelix
>
> Specifically, it refers to the dual-system model, corresponding to the closed Helix architecture of Figure AI. Our goal is to further analyze and reveal it in an academic context.

---

> ### Author Response · Authors · 2025-11-29
> **Response to the Reviewer Bsn4 (part 4)**
>
> ##### 2.5 About SOTA
>
> We would like to emphasize that the main objective of this paper is **not** to chase SOTA performance, but to provide **insights into how to design dual-system architectures**. Accordingly, it is more appropriate and informative to compare primarily against **other dual-system approaches**, rather than against the strongest single-system VLAs.
>
> Moreover, in order to ensure **experimental comparability and control**, our choices of VLM backbone and policy architecture are **intentionally not the most powerful ones available**. Under this setting, direct comparison with much stronger baselines would be inherently **unfair**, as performance differences would likely be dominated by backbone capacity rather than by the dual-system design itself.
>
> We fully understand your concern about using stronger baselines. However, we hope you can also appreciate that this research area is currently in a somewhat **chaotic and underspecified state**: many related works do not report sufficient implementation details, yet we still need to compare against them. As a result, we are effectively constrained to use relatively weaker but **reproducible** backbones, and introducing significantly stronger baselines at this stage would complicate the picture and make it harder to isolate the specific contribution of the **dual-system design**, which is the core focus of our work.
>
> In addition, our reliance on the **CALVIN** benchmark is largely dictated by the current landscape of **mainstream dual-system methods**: most prominent dual-system works (e.g., **LCB**, **RoboDual**) conduct their experiments on CALVIN, and the majority of these methods are **not open-sourced**. Consequently, at this moment we can only perform systematic experiments and comparisons on this benchmark. While this is not ideal, it is, in practice, the only feasible choice for ensuring a fair and consistent comparison with existing dual-system work.
>
> ##### 2.6 About writing
>
> Regarding the writing issues you mentioned, we have addressed each of them. Thank you again for your suggestions.

---

### Official Review · Reviewer_v8Uw · 2025-11-02

**Soundness:** 1
**Presentation:** 2
**Contribution:** 2
**Rating:** 4
**Confidence:** 4

**Summary:**

This paper investigates dual-system VLA architectures, where a high-level multimodal large language model (System 2) operates asynchronously with a low-level control policy (System 1). The authors present OpenHelix, an empirical framework analyzing different training strategies on the CALVIN benchmark.

**Strengths:**

The research problem is important: dual-system design addresses the critical trade-off between performance and efficiency in real-time robotic control.

**Weaknesses:**

- Classification of pseudo dual systems
   The paper classifies π₀ and GR00TN1 as pseudo dual systems because “their action experts do not receive real-time perceptual feedback.”
   However, according to their original papers:
   - π₀ operates in a *closed-loop* fashion, though without explicit asynchronous scheduling.
   - GR00TN1’s System 2 reportedly runs at around 10 Hz feedback frequency.
While these may not be fully asynchronous dual systems, this classification alone does not necessarily imply high time latency. To make the argument more rigorous, it would be helpful to include **quantitative latency and success rate** to justify dual system has low time latency while keep high performance. Simply assigning this label without testing might overlook the fact that some closed-loop single-system architectures already maintain reasonably low-latency feedback.

- The experiments would benefit from additional real-world validation.
  Since the paper’s motivation emphasizes *real-time deployment*, including results from a physical robot setup or latency profiling would make the findings more persuasive and practically relevant.

- The overall experimental design could be guided by a clearer unifying objective.  It remains somewhat unclear why the dual-system structure is fundamentally needed, and which specific components should be updated to achieve better performance.

- The comparison scope might be further expanded.
  In particular, Q1 could include stronger closed-loop single-system baselines such as **RobotVLM** or other online perception-action models, rather than focusing mainly on pseudo dual systems (RoboFlamingo, π₀, GR00TN1).

- The paper currently does not analyze time latency or efficiency trade-offs in depth.  Since asynchronous real-time performance is one of the main motivations, a latency comparison between single- and dual-system VLAs would significantly strengthen the empirical argument.

**Questions:**

On the claim that “without pre-alignment the model completely fails” This is a strong assertion.  Many recent VLM-based robotic policies (e.g., OpenVLA, Pi0) can be trained end-to-end without explicit pre-alignment modules.  Why does OpenHelix require pre-alignment so critically?  A deeper explanation or ablation could make this claim more convincing.

---

> ### Author Response · Authors · 2025-11-29
> **Response to the Reviewer v8Uw (part1)**
>
> #### 1.1. Classification of pseudo dual systems
>
> ##### 1.1.1 We first clarify our use of the term *dual-system* from a theoretical perspective.
>
> The notion of a dual-system architecture is originally grounded in cognitive neuroscience, where the two systems are defined with relatively clear functional boundaries. Dual-process theory posits that human cognition operates through two distinct systems:
>
> 1. **System 1** is fast, automatic, intuitive, and largely unconscious. It relies on heuristics, generates rapid responses, and is therefore efficient but also susceptible to systematic biases.
> 2. **System 2** is slow, deliberative, effortful, and conscious. It performs explicit reasoning, planning, and careful evaluation, but requires substantial cognitive resources.
> 3. The two systems operate in parallel and update information at different temporal frequencies.
>
> A crucial aspect of this formulation is that *both* System 1 and System 2 have direct access to perceptual input and can, in principle, produce actions concurrently. In other words, whether both systems simultaneously consume observation streams—rather than System 1 only operating on latent outputs from System 2—is central to the definition.
>
> From this perspective, a genuine dual-system architecture requires that both System 1 and System 2 are capable of making decisions and generating actions in parallel, each grounded directly in observations. This is precisely the distinction between Helix architecture of Figure  and the so‑called *pseudo dual systems* such as pi0 and GROOT, where the fast pathway effectively conditions only on latent embeddings or precomputed plans from the slow pathway, rather than on fresh perceptual input.
>
> ##### 1.1.2 We now explain the distinction from the standpoint of practical robotic control.
>
> The real-time performance (e.g., ~10 Hz) reported by prior *pseudo dual systems* (such as pi0/GROOT) is typically achieved via trunking or chunked inference. However, in these designs the “fast” execution still operates on *stale* observations: the policy acts on previously encoded visual input rather than continuously updated sensory streams.
>
> In the relatively simple, static scenarios used in these works—e.g., tabletop pick-and-place with mostly stationary objects—this limitation is often not apparent, because the robot arm can afford to “wait” for the model to finish its computation and the environment does not change significantly during inference. Once we move to more realistic and challenging settings, this becomes a critical failure mode. For example:
>
> 1. **Dynamic, fast-changing environments.** When objects or the robot itself move rapidly, actions derived from outdated observations can become invalid before execution completes. This leads to systematic divergence between the internal representation and the actual scene state.
> 2. **Contact-rich manipulation.** Tasks such as RAM insertion, cable routing, or folding deformable materials demand continuous, fine-grained adjustment during contact. Such behavior is impossible if the controller only reacts to delayed, asynchronous perceptual feedback.
>
> Therefore, although works such as pi0 and GROOT may appear dual-system-like under simple, quasi-static pick-and-place setups, they do *not* realize a true dual-system architecture in the sense described above, and they are fundamentally limited in many realistic manipulation scenarios that require high-bandwidth, closed-loop control.
>
> ##### 1.1.3 Relation to GEN-0 and motivation for our dual-system design
>
> Going one step further, GEN-0-style architectures explicitly emphasize *simultaneous* inference and control. The tasks demonstrated by GEN-0—for instance, folding cardboard—are contact-rich and require smooth, continuous correction in real time. Achieving this level of dexterity is only possible when perception and action are tightly coupled in a high-frequency closed loop.
>
> Whether we frame this in terms of dual-system theory or GEN-0-style implementations, the underlying motivation is the same: enabling more dexterous manipulation through continuous, real-time feedback and parallel control pathways.
>
> In contrast, *pseudo dual systems* primarily focus on **optimizing the mapping** from visual/language inputs to actions and obtaining additional performance gains via **action chunking** or macro-actions. While this yields efficiency benefits, the resulting execution remains driven by **asynchronous** observations. As a consequence, such systems inevitably fail on many real-world tasks that require fine-grained, online adjustment under rapidly changing conditions.
>
> This is precisely why we argue that exploring *true* dual-system architectures—where both fast and slow systems directly and concurrently consume observations and can each issue control—is necessary, beyond what pseudo dual systems can offer.

---

> ### Author Response · Authors · 2025-11-29
> **Response to the Reviewer v8Uw (part2)**
>
> #### 1.2. About real-world validation.
>
> Regarding real-robot experiments, due to time constraints we only conducted three types of tasks: **Lift, Push, and Stack**.
>
> Under standard settings, these tasks achieve very high success rates. Therefore, to specifically evaluate **semantic robustness**, we modified the language instructions. Concretely, the original commands:
>
> - “Lift the watermelon”
> - “Put the pepper in the bowl”
> - “Stack green bowl on yellow bowl”
>
> were replaced with more complex, **previously unseen** descriptions, such as:
>
> - “Grasp the green, striped spherical object.”
>
> The results are as follows. These experiments demonstrate that the **OpenHelix** architecture preserves strong **semantic generalization**, rather than overfitting to specific templated commands. This observation is consistent with our simulation findings and further confirms the **generalizability** of our method.
>
> | **Model** | **Lift** | **Put** | **Stack** | **Average** |
> | :-------- | :------- | :------ | :-------- | :---------- |
> | 3DDA      | 25       | 20      | 5         | 16.7        |
> | **Ours**  | **90**   | **74**  | **92**    | **85.3**    |
>
> #### 1.3. About  a clearer unifying objective.
>
> In fact, we have already explained our objectives clearly in Figure 1 and in Appendices A.1/A.2/A.3. Here, I would like to briefly summarize them again for you:
>
> We fully agree with the reviewer that a coherent experimental objective is crucial. However, as detailed in Appendix A.1 and Table 8, the current landscape of dual-system VLAs exhibits substantial design variability, making direct comparisons difficult without a systematic ablation study. Specifically: (1) System 2: LCB employs LLaVA, DP-VLA and RoboDual use OpenVLA, while HiRT relies on InstructBLIP. (2) System 1: Low-level policies range from 3D Diffusion Actors and Transformers to EfficientNet-based architectures and DiT models. (3)Latent Representation Variance: Methods differ in using specific <ACT> tokens, direct encoder outputs, or pooled representations.
>
> Given this divergence, comparing “Model A” to “Model B” is largely meaningless because nearly every component differs. Therefore, our goal, as outlined in Appendix A.2, is to disentangle these variables and establish a standardized evaluation paradigm. We explicitly standardize “Simulators & Tasks” and “Framework” (Conditions 1, 2, 3, and 7 in Appendix A.2) to isolate and assess the impact of different training strategies (Conditions 4, 5, and 6). This approach allows us to answer five core questions (Q1–Q5) and identify the optimal configuration for each module.
>
> This systematic analysis leads naturally to OpenHelix, a “simple yet effective” dual-system VLA. Importantly, OpenHelix is not an arbitrary combination of components but the empirical result of our unification process: it integrates the generalization capability of a Prompt-Tuned MLLM, the efficiency of a Fine-Tuned Pre-trained Policy, and the robustness of a Pre-aligned Projector.
>
> Overall, our unifying objective is tomove the field away from arbitrary model designs and toward a standardized, scientifically grounded approach.
>
>
> ##### 1.3.1 why the dual-system structure is fundamentally needed
>
> As mentioned in Section 1.1, “Classification of Pseudo Dual Systems,” the dual-system structure is fundamentally designed to address the requirements of tasks that demand more dexterous operations. If you have any questions or would like to discuss this further, please feel free to reach out to us at any time.

---

> ### Author Response · Authors · 2025-11-29
> **Response to the Reviewer v8Uw (part3)**
>
> ##### 1.3.2 which specific components should be updated to achieve better performance
>
> Our paper explicitly dedicates Sections 5.2, 5.3, and 5.4 to answering this exact question. By isolating the high-level MLLM, the low-level Policy, and the connecting Projector, we determined the optimal update strategy for each component:
>
> High-Level MLLM (Update Strategy: Prompt Tuning): We found that fully updating (fine-tuning) the MLLM can degrade its inherent language generalization capabilities. Instead, the most effective strategy is to freeze the MLLM backbone and only update a learnable <ACT> token/prompt. Our experiments in Table 3 show that this strategy preserves generalization significantly better than full fine-tuning on CALVIN-E.
>
> Low-Level Policy (Update Strategy: Fine-Tuning from Pre-trained): For the policy component, we compared training from scratch versus fine-tuning. Our results in Table 4 demonstrate that updating a pre-trained policy yields superior performance compared to training from scratch, as it leverages prior motion priors for faster convergence.
>
> Connection Layer (Update Strategy: Pre-aligned Projector): We identified that the projector bridging the two systems must be updated with care. Table 5 shows that if the projector is trained jointly without pre-alignment, the model fails completely. Therefore, the specific update requirement here is to pre-align the projector while keeping other components frozen before joint optimization.
>
> In summary, our empirical analysis concludes that to achieve the best performance, one should prompt-tune the high-level MLLM while fine-tuning the low-level pre-trained policy, connected by a pre-aligned projector .
>
> #### 1.4. About a stronger baseline
>
> Thank you for your comment. As our paper shows, existing dual-system approaches vary widely in their design choices (e.g., different backbones, input modalities, policy structures, etc.). Our goal is not to compare against stronger baselines whose configurations differ along so many dimensions. These differences introduce significant confounding factors that make such comparisons uninformative.
>
> Instead, our intention is to clarify the current know-how of dual-system designs for embodied intelligence within a controlled and comparable setting. This allows us to isolate the core principles without being affected by implementation discrepancies across prior work.
>
> Of course, we can provide π₀ results if absolutely needed, but we believe such a comparison is unnecessary for the reasons outlined above.
>
> | **Success Rates** | 1_st task | 2_nd task | 3_th task | 4_th task | 5_th tasks | Num Task Success ↑ |
> | ----------------- | --------- | --------- | --------- | --------- | ---------- | ------------------ |
> | Openvla           | 73.1%     | 42.4%     | 24.0%     | 12.9%     | 7.5%       | 1.60               |
> | pi0               | 91.6%     | 82.1%     | 71.7%     | 64.1%     | 53.8%      | 3.63               |
> | Ours              | 97.1%     | 91.4%     | 82.8%     | 72.6%     | 64.1%      | 4.08               |
>
> #### 1.5. About time latency
>
> We strongly agree with your general point, but there is one important nuance: **in simulation**, such experiments typically **do not include time latency**, since the simulator executes actions almost instantaneously.
>
> Under our current setup, we achieve an inference rate of roughly **10 Hz**. However, this is largely because the downstream policy is a **3D-based controller**, whose processing is relatively expensive. To better understand this effect, we conducted an ablation in which we replaced the downstream policy with a **2D Flow-Matching head**. Under this setting, we observed a **2–3× speedup** in inference, with a slight **improvement in performance**.
>
> That said, we believe that a **raw latency comparison between single- and dual-system VLAs** is **not particularly meaningful**. Latency is heavily influenced by implementation details such as:
>
> - the specific **VLM architecture** used, and
> - the **design of the downstream policy network**.
>
> These factors make it very difficult to ensure a fair, apples-to-apples comparison purely at the architectural level.
>
> If the concern is more about the **intrinsic value of the dual-system architecture itself**, we would refer you to Section **2.1.2, “Why pseudo dual-system VLAs necessarily fail in dynamic tasks”**, where we explain in detail why a true dual-system design is needed for dynamic environments, beyond what can be resolved by simply measuring or optimizing latency.

---

> ### Author Response · Authors · 2025-11-29
> **Response to the Reviewer v8Uw (part4)**
>
> #### 1.6 About "why pre-alignment so critically"
>
> First, the necessity of **pre-alignment** arises from an inherent **objective mismatch**: the optimization goal for the vision–language (VL) to action mapping is not the same as the optimization goal for the action generator / downstream policy. These are not a single unified objective at the beginning.
>
> Our procedure is therefore structured in two stages:
>
> 1. **Pre-alignment stage (VL → A mapping).**
>    We first freeze both the upstream VL encoder and the downstream policy, and introduce a lightweight projector on top of the upstream latent features to perform a domain adaptation step. The goal is to map the upstream latent space into a representation that the downstream policy can partially interpret. Conceptually, this stage is learning the VL-to-action *interface*: it aligns the upstream latent features to the representation manifold expected by the downstream policy.
> 2. **Joint optimization stage (true dual-system training).**
>    After this pre-alignment, we unfreeze both sides and perform joint optimization of the dual system. At this point, the two systems are no longer trying to solve completely incompatible objectives from scratch; instead, the upstream representations have already been nudged into a regime where the downstream policy can meaningfully consume them.
>
> One subtle but crucial detail—which may be easy to overlook—is that the downstream policy itself is **already pre-trained**, and in our design we are **replacing one of its original conditioning inputs** with the latent features produced by the upstream system.
>
> The downstream conditions are not arbitrary: they live in a **constrained, structured space** that is compatible with the rest of the policy’s inputs and internal representations. Consequently, the upstream latent features cannot be freely optimized; they must be **aligned into this constrained conditioning space** of the downstream policy. This makes the optimization problem fundamentally different from training a single monolithic system end-to-end from scratch, where all representations can co-adapt jointly without such constraints.
>
> This mechanism is conceptually analogous to why the original LCB [1] paper required the use of a **CLIP loss** to work properly: in both cases, a pre-existing representation space imposes structure on how new components must be aligned, and an explicit alignment objective (or pre-alignment stage) is essential for successful integration.
>
> [1] LCB: Latent Codes as Bridges in Hierarchical Robot Control

---

### Meta-Review · Area_Chair_781J · 2026-01-02

**Summary:**

Based on the reviewer feedback and the concerns, summarized below, I recommend rejection at this time.

**v8Uw** states that the *strength* is the timeliness of studying dual-system designs in VLAs. The main *concerns* are (1) the need to quantify the latency and success rate to justify that a dual system has low time latency while keep high performance, (2) real-world evaluation, (3) clearer evaluation, baselines, and ablations to justify the need for a dual-design framework, (4) why does openhelix need pre-alignment when other SOTA VLAs do not?

**Bsn4** states that the *strengths* are the introduction of two new evaluation suites, the paper is generally well-written, the paper systematically evaluates the components, and has a large number of empirical tests. The main *concerns* are about (1) details of evaluation and baselines in Section 5.1, (2) the generalizability of prompt tuning results, (3) fine-tuning outperforming training from scratch in Section 5.3 is not a novel finding, (4) need for a clearer and more precise definition of what the openhelix system is, (5) the need for more benchmarks beyond CALVIN.

**9bia** states that the *strengths* are new benchmark tasks and systematic evaluation of the simple openhelix pipeline, with empirical results showing favorable performance. The main *concerns* are (1) lack of real-world experiments, (2) limited discussion on the low-level control strategies and their generalizability, (3) need for more details on training data and optimization strategies for fair comparison.

**oK3Z** states that the *strengths* are very thorough empirical evaluation and ablations, new benchmark domains, and strong results over baselines. The main *concerns* are (1) the methodological contribution (OpenHelix) mainly aggregates established best practices, (2) simulation only experiments, (3) need for more investigation on asynchronous inference, (4) DSVLABench overlaps with prior frameworks like RoboBench or RT-X.

Some additional papers that are relevant and I would recommend citing and discussing:
* [1] Lin, Fanqi, et al. "OneTwoVLA: A Unified Vision-Language-Action Model with Adaptive Reasoning." arXiv preprint arXiv:2505.11917 (2025).
* [2] Liu, Jiaming, et al. "Hybridvla: Collaborative diffusion and autoregression in a unified vision-language-action model." arXiv preprint arXiv:2503.10631 (2025).

**Reviewer Concerns:**

**v8Uw Concerns**
* (1) the need to quantify the latency and success rate to justify that a dual system has low time latency while keep high performance $\rightarrow$ **partially addressed (discussion)**. The rebuttal justifies the “term dual-system from a theoretical perspective”, stating the colloquially well-establishing distinction of system 1 / system 2 as fast vs. slow reactions or thinking. The rebuttal also attempts to discuss how these intuitive definitions are mapped practically into a distinction, but I found the definitions to still be too hand-wavey. The work would be significantly strengthened if it provided a rigorous definition of what system 1 / system 2 thinking should even mean in robotics (e.g., what frequency do each of these models have to operate on to be considered distinct? Or is system 1 a feedback policy while system 2 a model predictive planner?, etc.). Much of the discussion in the rebuttal should be formalized and distilled into the main paper in order to solidify the key contribution of this paper, which is the dual control framework and evaluation itself.
* (2) real-world evaluation $\rightarrow$ **partially addressed**. Authors seem to have added three real-world tasks: Lift, Push, and Stack; although it is hard to know the details of the real-world eval like what the robot platform is, what the camera setups were, etc.
* (3) clearer evaluation, baselines, and ablations to justify the need for a dual-design framework $\rightarrow$ **partially addressed**.
* (4) why does openhelix need pre-alignment when other SOTA VLAs do not? $\rightarrow$ **addressed**.

**Bsn4 Concerns**
* (1) details of evaluation and baselines in Section 5.1 $\rightarrow$ **addressed (discussion)**
* (2) the generalizability of prompt tuning results $\rightarrow$ **partially addressed**
* (3) fine-tuning outperforming training from scratch in Section 5.3 is not a novel finding $\rightarrow$ **partially addressed (discussion)**. It was hard to parse the argument presented in the rebuttal on why the fine-tuning findings are somehow novel or different compared to what the rest of the VLA community has found so far. For example, what does “this model itself is an independent VLA-capable policy” mean?
* (4) need for a clearer and more precise definition of what the openhelix system is, (5) the need for more benchmarks beyond CALVIN $\rightarrow$ **partially addressed**

**9bia Concerns**
* (1) lack of real-world experiments, $\rightarrow$ **partially addressed**. Authors seem to have added three real-world tasks: Lift, Push, and Stack; although it is hard to know the details of the real-world eval like what the robot platform is, what the camera setups were, etc.
* (2) limited discussion on the low-level control strategies and their generalizability, $\rightarrow$ **partially addressed**. Discussion about how the low-level policy should be 3DDA but they also implemented 2D in the original setup. It was hard to tell if this addressed the generalizability aspect of the question however.
* (3) need for more details on training data and optimization strategies for fair comparison. **addressed**. Discussion about how prior works have studied optimization tricks and they use standard benchmark training data.

**oK3Z Concerns**
* (1) the methodological contribution (OpenHelix) mainly aggregates established best practices,  $\rightarrow$ **not addressed**. The authors acknowledge this point but do not elaborate much on it.
* (2) simulation only experiments $\rightarrow$ **partially addressed**. Authors seem to have added three real-world tasks: Lift, Push, and Stack; although it is hard to know the details of the real-world eval like what the robot platform is, what the camera setups were, etc.
* (3) need for more investigation on asynchronous inference $\rightarrow$ **partially addressed (discussion, somewhat roundabout)**.
* (4) DSVLABench overlaps with prior frameworks like RoboBench or RT-X $\rightarrow$ **partially addressed**.

**Reviewer Scores:**

* **v8Uw** would have *maintained* a score of 4: marginally below the acceptance threshold.
* **Bsn4** would have *maintained* a score of 4: marginally below the acceptance threshold.
* **9bia** would have *maintained* a score of 4: marginally below the acceptance threshold.
* **oK3Z** would have *reduced* to a score of 6: marginally above the acceptance threshold.

---

### Decision · Program_Chairs · 2026-01-26

Reject